# WeatherPrompt: Multi-modality Representation Learning for All-Weather Drone Visual Geo-Localization

**Jiahao Wen**[1]    **Hang Yu**[1][*]   **Zhedong Zheng**[2]

[1]School of Computer Engineering and Science, Shanghai University, China
[2]Faculty of Science and Technology and Institute of Collaborative Innovation, University of Macau, China
{wenjh,yuhang}@shu.edu.cn, zhedongzheng@um.edu.mo

## Abstract

Visual geo-localization for drones faces critical degradation under weather perturbations, *e.g.*, rain and fog, where existing methods struggle with two inherent limitations: 1) Heavy reliance on limited weather categories that constrain generalization, and 2) Suboptimal disentanglement of entangled scene-weather features through pseudo weather categories. We present WeatherPrompt, a multi-modality learning paradigm that establishes weather-invariant representations through fusing the image embedding with the text context. Our framework introduces two key contributions: First, a Training-free Weather Reasoning mechanism that employs off-the-shelf large multi-modality models to synthesize multi-weather textual descriptions through human-like reasoning. It improves the scalability to unseen or complex weather, and could reflect different weather strength. Second, to better disentangle the scene and weather features, we propose a multi-modality framework with the dynamic gating mechanism driven by the text embedding to adaptively reweight and fuse visual features across modalities. The framework is further optimized by the cross-modal objectives, including image-text contrastive learning and image-text matching, which maps the same scene with different weather conditions closer in the representation space. Extensive experiments validate that, under diverse weather conditions, our method achieves competitive recall rates compared to state-of-the-art drone geo-localization methods. Notably, it improves Recall@1 by 13.37% under night conditions and by 18.69% under fog and snow conditions. Our code is available at https://github.com/Jahawn-Wen/WeatherPrompt.

## 1 Introduction

Drone visual geo-localization aims to match drone-view image with corresponding satellite views, supporting critical applications such as disaster response, urban surveillance, search-and-rescue, and environmental monitoring [1, 2, 3, 4, 5, 6]. However, variable weather conditions such as rain, fog and snow introduce noise, occlusions and low visibility, which severely distort image features [7, 8, 9] and leading conventional localization methods to suffer drastic performance degradation under extreme weather. Recent advances in cross-modal retrieval show that integrating natural language descriptions can substantially enhance the discrimination power of vision models, allowing better generalization in complex or ambiguous scenarios [10, 11, 12, 13]. Despite this progress, leveraging textual guidance for cross-weather drone geo-localization remains largely underexplored, especially considering the nuanced and dynamic nature of weather conditions encountered in the field. The ability of text to capture complex semantics and fine-grained details [14, 15] offers a promising

---

[*]Corresponding author.

39th Conference on Neural Information Processing Systems (NeurIPS 2025).

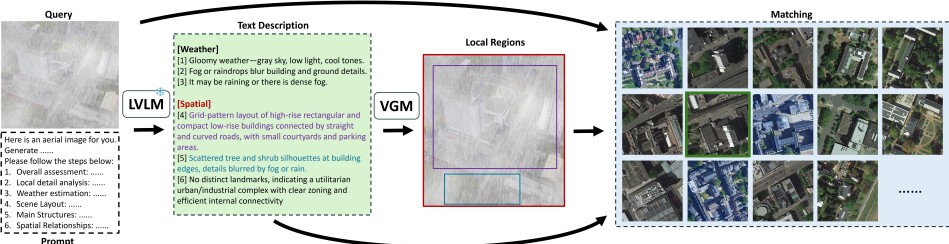

Figure 1: **Example of the proposed Chain-of-Thought description and matching.** Our framework generates structured weather and spatial Text Description via stepwise reasoning. We leverage Off-the-shelf Visual Grounding Model (VGM), *i.e.*, XVLM [16] to extract local region cues, which are integrated to further refine the matching process. Finally, we match images using weather description, global scene layout, and local region semantics to retrieve the corresponding satellite-view image.

avenue for cross-weather generalization. Addressing the all-weather visual geo-localization task presents two primary challenges: (1) Limited Weather Labeling. Existing approaches typically perform domain-specific fine-tuning on a limited set of predefined weather labels (*e.g.*, sunny, rainy). This closed-set paradigm fails to capture the continuous and combinatorial nature of real-world weather, thereby limiting model generalization to unseen or mixed conditions and preventing the exploitation of richer weather semantics. (2) Scene–Weather Feature Entanglement. Existing methods directly inject coarse pseudo-weather labels (*e.g.*, rain, fog) into visual representations during training, leading to a severe entanglement between scene semantics and weather disturbances. Consequently, the model learns suboptimal representations under mixed or unseen weather conditions, fails to disentangle scene content from weather noise, and is severely limited in cross-weather generalization.

For the first limitation, we propose a training-free weather reasoning mechanism leveraging off-the-shelf large multimodal models [17] (see Fig. 1). Specifically, we employ chain-of-thought (CoT) [18] prompting to automatically generate rich, step-by-step natural language weather descriptions for each geographical scene using a single randomly sampled drone-view image. This strategy circumvents manual description and expert controlled, enabling large-scale collection of high-quality and diverse multimodal samples at significantly reduced costs. Moreover, the stepwise reasoning introduced by the CoT prompts ensures semantic accuracy and formatting consistency across generated captions, further enhancing the reliability and usability of descriptions. This ultimately leads to improved generalization for complex and unseen weather conditions. To address the second challenge, we propose a multimodal framework equipped with a text embedding-driven dynamic gating mechanism to adaptively reweight and fuse visual features, effectively disentangling scene and weather attributes. Specifically, during training, the framework jointly optimizes multimodal objectives including image-text contrastive (ITC) loss and image-text matching (ITM) loss, aligning drone images with generated weather-aware captions. Additionally, a localized alignment loss is introduced to explicitly enforce consistency between annotated visual regions and textual descriptions across multiple granularities, encouraging the visual encoder to learn robust and weather-invariant scene representations. At inference, visual and textual embeddings are extracted in parallel, and the textual embeddings dynamically modulate visual features via the gating mechanism. The resulting multimodal representation is directly fed into a classification head for localization prediction, avoiding any additional online fine-tuning or dedicated parameter sets for different weather conditions. This design significantly reduces structural complexity and improves the deployment efficiency on resource-constrained platforms. The main contributions are as follows:

- **Training-Free Weather Reasoning**: We pioneer automatic weather semantics extraction through Large Vision Language Models (LVLMs) with chain-of-thought prompting, eliminating manual descriptions. Our hierarchical reasoning mechanism integrates continuous weather priors with spatial-object analysis, enabling weather-adaptive caption generation and scalable dataset construction.

- **Semantic Disentanglement via Language Guidance**: We devise a text-driven framework achieving scene-weather disentanglement through: (1) Multi-granularity alignment of visual features with continuous textual weather embeddings, (2) Region-level semantic consistency enforcement, (3) Dynamic textual gating for weather-invariant representations.

- **State-of-the-Art Generalization**: The proposed method has achieved average 87.72% Recall@1 on University-1652 and 83.1% on SUS-200 over 10 different weather conditions. For unseen weather combinations (*i.e.*, Dark+Rain+Fog), ours still arrives at 72.15% AP, validating unprecedented cross-domain generalization.

## 2  Related Work

**Cross-view Geolocalization.** Cross-view geo-localization aims to match images captured from different viewpoints with their corresponding geographic locations [1, 19, 20, 21]. Early approaches relied on hand-crafted local features such as SIFT [22] and SURF [23], as well as global descriptors like VLAD [24] and Fisher Vector [25], often combined with RANSAC [26] for geometric verification; however, they remain brittle under large viewpoint and illumination changes [27, 28]. With the advent of deep learning, deep metric learning frameworks based on global or part-based contrastive objectives have become dominant [29, 30, 31, 32, 33, 34]. These approaches employ triplet or InfoNCE losses to train end-to-end embeddings and integrate global pooling with spatial attention or multi-region partitioning strategies [35, 36]. Representative examples include the multi-part partitioning scheme *et al.* [2], the keypoint attention module [3], the content-aligned Transformer architecture [37], self-attention positional encoding by Yang *et al.* [35], the dual-path fusion network [36], and Bird's Eye View (BEV) [38], all of which substantially enhance cross-view feature alignment. Recent research has begun to address the impact of image degradations such as low-light, motion blur, and synthetic fog, often by using data augmentation, domain adaptation, or cross-modal transformers [39, 40, 41, 42, 43, 44]. However, most existing methods still depend on a limited set of discrete weather labels, restricting generalization to unseen or complex conditions. In contrast, our approach introduces a training-free, all-weather text-guided representation learning framework that leverages open-set weather descriptions to overcome these limitations.

**Multi-modality Alignment.** In this work, we address weather-aware text-guided representation learning, where the goal is to retrieve drone-view images based on fine-grained weather-related textual cues. Recent advances in vision–language alignment, such as CLIP [45], BLIP [46], and XVLM [16], have established powerful contrastive pre-training and cross-modal attention mechanisms, but previous studies mainly target static semantics or rigid spatial relations [47, 48, 49]. Additional efforts on adaptive fusion and region-word alignment [50, 51, 52, 53] have improved retrieval, but remain limited by closed vocabularies and overlook dynamic, fine-grained weather semantics. To address these gaps, we propose a framework that generates open-set weather descriptions via Chain-of-Thought prompting and applies text-driven dynamic gating for adaptive feature modulation, achieving robust cross-modal alignment under diverse weather conditions.

**Large Vision Language Models for Vision via Prompting.** Large Vision Language Models (LVLMs) such as GPT-3/4 [54, 55] and Qwen [17] have recently been applied to vision tasks using prompt engineering. Approaches like VisualGPT [56] and MM-CoT [57] leverage Chain-of-Thought (CoT) prompts [18] to elicit stepwise reasoning for visual question answering and captioning. However, scaling LVLMs to cross-view, multi-weather geo-localization remains challenging: free-form text often suffers from hallucinations and lacks semantic or structural consistency [58, 59, 60]. Prior works also overlook prompt design tailored for robust, multi-weather, multi-scale cross-modal alignment. To address this, we introduce the first CoT-driven description pipeline for multi-weather drone-to-satellite geo-localization. Our structured prompts regularize LVLMs' outputs, enabling scalable generation of high-quality, open-set weather descriptions to advance large-scale vision–language alignment.

## 3  Method

### 3.1  Open-Weather Description

As shown in Fig. 2, we present an overview of our multi-weather drone-view image captioning pipeline. To minimize description redundancy and mitigate overfitting, our approach begins by randomly sampling a single representative drone-view image from each geographical region, forming concise yet diverse image–text pairs. Notably, Large Vision Language Models (LVLMs) typically generate weather labels or captions intuitively, which may omit critical visual cues and exhibit semantic inconsistency. To address this limitation, we explicitly divide the captioning process into

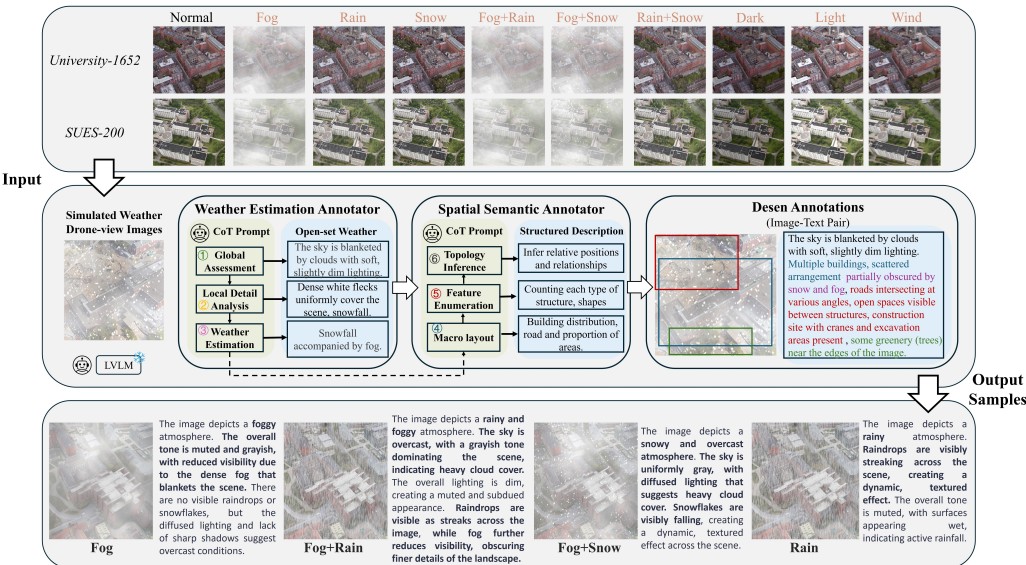

Figure 2: **The proposed training-free weather reasoning mechanism**. We synthesize drone-view images with diverse weather conditions based on the University-1652 and SUES-200 datasets, covering complex scenarios such as fog, rain, snow, and nighttime. For each synthesized image, we first employ stepwise Chain-of-Thought prompting to generate open-set weather descriptions, including global assessment, local detail analysis, and weather inference. Guided by the inferred weather prior, we then sequentially reason about the scene's macro layout, structural elements, and topological relationships, ultimately producing high-quality, structured image–text pairs.

two sequential phases: a weather estimation phase and a spatial semantics phase, guided by carefully designed Chain-of-Thought (CoT) prompts. Inspired by human hierarchical visual reasoning, our CoT prompting strategy enforces a rigorous three-step reasoning procedure: global perception, local analysis, and comprehensive synthesis.

In the weather estimation phase, the LVLMs first assesses global visibility to quantify observable range, subsequently identifies localized atmospheric indicators such as rain-streak reflections or fog diffusion patterns, and finally integrates these global and local cues to assign an accurate weather category. This structured inference process significantly alleviates the ambiguity inherent to single-shot captioning methods, establishing a reliable weather-conditioned prior for the following stage.

In the subsequent spatial semantics phase, conditioned explicitly on the inferred weather semantic, the model rapidly identifies macro-level layout features, including the spatial distribution of buildings, orientations of roads, and proportion of open spaces. Then, it captures micro-level structural details such as object counts, shapes, and local spatial arrangements. Ultimately, the pipeline synthesizes these cues by reasoning about relative positions and topological relationships to generate a structured textual description. This final output aligns comprehensively with weather semantics, spatial structure, and detailed scene attributes, ensuring robust, consistent, and high-quality multimodal descriptions.

**Single-image Sampling.** Since existing benchmarks (*e.g.*, University-1652) are partitioned by geographic regions, we randomly select only one drone-view image per region as a representative to avoid redundancy. Given the prohibitive cost of obtaining real drone-view imagery across diverse weather conditions, we utilize the imgaug [61] library to synthesize realistic weather variations. By parametrically adjusting visibility and occlusion effects such as rain, fog, and snow, we generate high-fidelity, diverse meteorological scenarios that closely resemble real-world conditions.

**Weather Estimation Phase.** Given a drone-view image with synthetically generated weather, we apply a pretrained large multimodal model [17] for automatic weather description. To mitigate hallucinations, vague phrasing, and inconsistent terminology typical of single-shot captioning, we adopt a Chain-of-Thought prompting strategy inspired by human visual perception. Specifically, we first quantify global visibility, then identify local meteorological cues such as rain-streak reflections or fog diffusion, and finally integrate global and local evidence to determine a reliable weather label. Any description that lacks explicit visibility information, meteorological cues, or contains

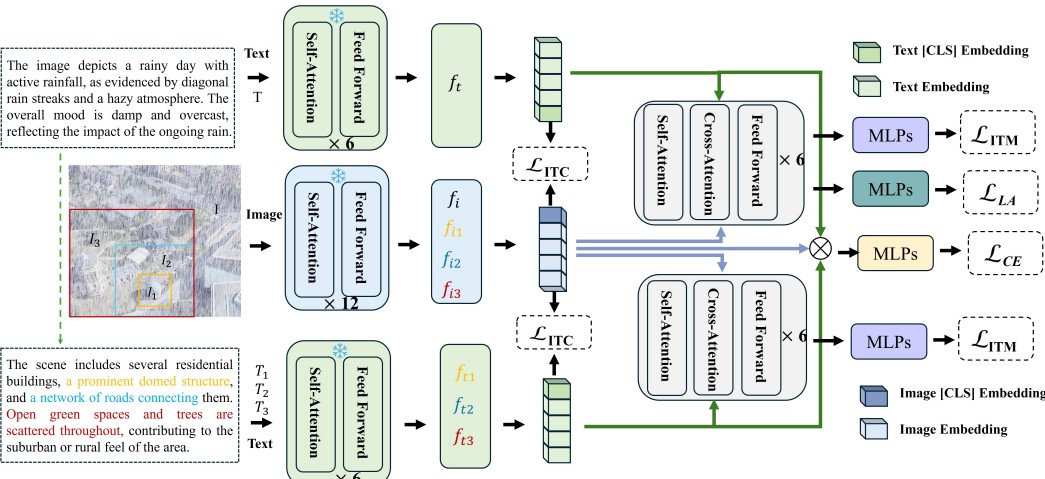

Figure 3: **The proposed multimodal alignment framework.** Our model extracts global and local features from drone images and multi-step weather captions, performs multi-granular image-text alignment, and dynamically fuses modalities via weather-driven gating for robust geo-localization.

uncertain terminology (*e.g.*, "possibly" or "uncertain") is automatically rejected and regenerated. This three-step reasoning pipeline ensures accurate, consistent, and reliable weather descriptions, thus providing a robust prior for the subsequent spatial semantics phase.

**Spatial Semantics Phase.** Conditioned on the inferred weather prior, we further generate detailed, scene-level textual descriptions essential for precise vision–language alignment. Initially, the model conducts a global scan to capture macro-level structures, including building distributions, road orientations, and proportions of open areas. Subsequently, it enumerates fine-grained elements, accurately counting structural entities, describing shapes, and detailing local spatial arrangements. Lastly, it synthesizes macro-layout and micro-level information to explicitly infer relative positions and topological relationships, producing structured, natural-language scene descriptions. These descriptions serve as fine-grained semantic supervision for downstream multimodal alignment.

**Discussion.** Existing cross-view geo-localization methods primarily rely on image-based retrieval. While some recent studies [21, 62] integrate textual descriptions, these approaches remain limited to coarse, scene-level labels and largely neglect weather semantics, resulting in compromised robustness under diverse weather conditions. In contrast, we introduce a training-free, CoT-driven pipeline for automatically generating multi-weather semantic descriptions. By employing large multimodal models guided by Chain-of-Thought prompting, we move beyond predefined discrete weather categories and achieve robust generalization to unseen and complex meteorological conditions. Furthermore, our framework systematically annotates both macro-level spatial layouts and micro-level structural attributes, embedding essential spatial cues into the image–text pairs. Relying solely on pretrained multimodal models without manual description, our approach efficiently produces large-scale, semantically consistent multimodal data, significantly facilitating downstream cross-weather multimodal representation learning.

## 3.2 Multimodal Alignment Model

We introduce a training-free, all-weather text-guided representation learning framework (see Fig. 3) that aligns weather semantics injected via text prompts with visual features to produce consistent, discriminative cross-view embeddings across diverse meteorological conditions. The framework consists of a pretrained visual encoder, a text encoder, and a cross-modal fusion module, capped by a lightweight classification head. During inference, the text embedding parametrizes a gating mechanism that adaptively modulates the visual feature channels; the fused multimodal representation is then passed through a single-layer MLP to predict the geographic location.

**Weather-Driven Channel Gating.** Let $f_I \in \mathbb{R}^{B \times D}$ and $f_T \in \mathbb{R}^{B \times D}$ represent the normalized visual and textual embeddings for a batch of $B$ samples. We utilize text embeddings to generate an adaptive

channel-wise gating vector $g \in (0,1)^{B \times D}$:

$$z = \text{ReLU}\big(f_{\text{T}} W_1^{(gate)\top} + b_1^{(gate)}\big) \in \mathbb{R}^{B \times D/r}, \quad g = \sigma\big(z W_2^{(gate)\top} + b_2^{(gate)}\big) \in (0,1)^{B \times D}, \tag{1}$$

where $W_1^{(gate)}, W_2^{(gate)}, b_1^{(gate)}, b_2^{(gate)}$ are learnable parameters, and $r$ is the reduction ratio. The fused multimodal feature is computed by

$$f_{\text{fuse}} = g \odot f_{\text{I}} + (1 - g) \odot f_{\text{T}} \quad \in \mathbb{R}^{B \times D}, \tag{2}$$

followed by a classification head for geo-localization. By dynamically modulating visual channels conditioned on textual weather semantics, our approach yields discriminative and weather-robust embeddings with minimal computational overhead.

**Image-Text Contrastive.** Given paired samples $\{(I_i, T_i)\}_{i=1}^{B}$, we compute the similarity matrix $S_{ij} = \frac{I_i^\top T_j}{\tau} \in \mathbb{R}^{B \times B}$ between normalized visual embeddings $I_i$ and textual embeddings $T_j$, where $\tau$ is a learnable temperature parameter. We then convert the diagonal entries into retrieval probabilities by applying softmax over rows and columns:

$$p_{I \to T}^{(i)} = \frac{\exp(S_{ii})}{\sum_{j=1}^{B} \exp(S_{ij})}, \quad p_{T \to I}^{(i)} = \frac{\exp(S_{ii})}{\sum_{j=1}^{B} \exp(S_{ji})}. \tag{3}$$

The contrastive loss is defined as:

$$\mathcal{L}_{\text{ITC}} = -\frac{1}{2B} \sum_{i=1}^{B} \Big[ \log p_{I \to T}^{(i)} + \log p_{T \to I}^{(i)} \Big], \tag{4}$$

enforcing global semantic alignment between the visual and textual modalities.

**Image-Text Matching.** To further enhance fine-grained discriminability, we introduce an image-text matching loss utilizing hard negatives. Given the similarity matrix $S$, for each image $I_i$ and its paired textual $T_i$), we select the highest non–diagonal similarity entry as its hard negative: $T_i^- = \arg\max_{j \neq i} S_{ij}, I_i^- = \arg\max_{j \neq i} S_{ji}$. This forms $2B$ hard negative pairs $(I_i, T_i^-)$ and $(I_i^-, T_i)$, and $B$ positive pairs $(I_i, T_i)$, resulting in $3B$ pairs in total. Each pair is passed through the cross-modal encoder, and we extract the CLS embedding $h_k$ which is fed to a binary classifier $f_{\text{match}}$. The matching probability is $p_k = \sigma\big(f_{\text{match}}(h_k)\big)$, where $\sigma$ is the sigmoid function. Denoting the ground-truth label $y_k = 1$ for positive pairs and $y_k = 0$ for negatives, the matching loss is defined as

$$\mathcal{L}_{\text{ITM}} = -\frac{1}{3B} \sum_{k=1}^{3B} \Big[ y_k \log p_k + (1 - y_k) \log(1 - p_k) \Big], \tag{5}$$

which guides the model to hear subtle semantic distinctions between closely related pairs.

**Localized Alignment Module.** To enable text-driven fine-grained visual localization, the model responds to both global descriptions $T$ and the region hints $T_1, T_2, T_3$. For the j-th concept , we extract its [CLS] embedding $x_{\text{cls}}^j \in \mathbb{R}^d$. from the cross-encoder, which is projected to normalized coordinate vector via a dedicated two-layer MLP:

$$\hat{l}^j = \sigma\big(W_2^{(loc)} \text{GELU}(W_1^{(loc)} x_{\text{cls}}^j + b_1^{(loc)}) + b_2^{(loc)}\big) \in [0,1]^4 \tag{6}$$

where $W_1^{(loc)} \in \mathbb{R}^{2d \times d}$, $W_2^{(loc)} \in \mathbb{R}^{4 \times 2d}$, $b_1^{(loc)} \in \mathbb{R}^{2d}$, $b_2^{(loc)} \in \mathbb{R}^4$. The resulting vector $\hat{l}^j = (\hat{c}_x, \hat{c}_y, \hat{w}, \hat{h})$ encodes the region center and size in normalized coordinates. The localized alignment loss supervises both overlap and regression accuracy:

$$\mathcal{L}_{\text{LA}} = \mathbb{E}_{(I,T^j) \sim \mathcal{D}} \Big[ \underbrace{1 - \text{IoU}\big(l^j, \hat{l}^j\big)}_{\text{Overlap Loss}} + \underbrace{\|l^j - \hat{l}^j\|_1}_{\text{L1 Loss}} \Big], \tag{7}$$

where $l^j = (c_x, c_y, w, h)$ is the ground-truth region, $\hat{l}^j$ its prediction, and $\text{IoU}(\cdot, \cdot)$ computes the Intersection-over-Union.

**Classification Module.** After multimodal fusion, we employ a single-layer MLP classification head to predict the geographic location. Let the fused feature for sample $b$ be $z_b \in \mathbb{R}^D$, and classifier

| Method | Normal | | Fog | | Rain | | Snow | | Fog+Rain | | Fog+Snow | | Rain+Snow | | Dark | | Over-exp | | Wind | | Mean | |
|---|---|---|---|---|---|---|---|---|---|---|---|---|---|---|---|---|---|---|---|---|---|---|
| | R@1 | AP | R@1 | AP | R@1 | AP | R@1 | AP | R@1 | AP | R@1 | AP | R@1 | AP | R@1 | AP | R@1 | AP | R@1 | AP | R@1 | AP |
| Drone → Satellite | | | | | | | | | | | | | | | | | | | | | | |
| Zheng et al.[1] [backbone] | 67.83 | 71.74 | 60.97 | 65.23 | 60.29 | 64.61 | 55.58 | 60.09 | 54.75 | 59.40 | 44.85 | 49.78 | 57.61 | 62.03 | 39.70 | 44.65 | 51.85 | 56.75 | 58.28 | 62.83 | 55.17 | 59.71 |
| ResNet-101 [68] [backbone] | 70.07 | 73.04 | 63.87 | 68.22 | 63.34 | 67.59 | 59.75 | 64.15 | 57.45 | 62.12 | 48.31 | 53.28 | 60.25 | 64.68 | 46.12 | 51.02 | 56.34 | 61.23 | 62.13 | 66.63 | 58.76 | 63.29 |
| DenseNet121 [69] [backbone] | 69.48 | 73.26 | 64.25 | 68.47 | 63.47 | 67.64 | 59.29 | 63.70 | 59.68 | 64.13 | 50.41 | 55.20 | 60.21 | 64.57 | 48.57 | 53.41 | 54.04 | 58.88 | 60.74 | 65.14 | 59.01 | 63.44 |
| Swin-T [64] [backbone] | 69.27 | 73.18 | 66.46 | 70.52 | 65.44 | 69.60 | 61.79 | 66.23 | 63.96 | 68.21 | 56.44 | 61.07 | 62.68 | 67.02 | 50.27 | 55.18 | 55.46 | 60.29 | 63.81 | 68.17 | 61.56 | 65.95 |
| IBN-Net [70] [backbone] | 72.35 | 75.85 | 66.68 | 70.64 | 67.95 | 71.73 | 62.77 | 66.85 | 62.64 | 66.84 | 51.09 | 55.79 | 64.07 | 68.13 | 50.72 | 55.53 | 57.97 | 62.52 | 66.73 | 70.68 | 62.30 | 66.46 |
| LPN [2] [TCSVT'21] | 74.33 | 77.60 | 69.31 | 72.95 | 67.96 | 71.72 | 64.90 | 68.85 | 64.51 | 68.52 | 54.16 | 58.73 | 65.38 | 69.29 | 53.68 | 58.10 | 60.90 | 65.27 | 66.46 | 70.35 | 64.16 | 68.14 |
| Sample4Geo* [71] [ICCV'23] | 92.70 | 93.85 | 88.70 | 90.55 | 62.44 | 66.17 | 52.76 | 57.24 | 52.70 | 56.77 | 19.79 | 23.16 | 38.19 | 42.33 | 46.34 | 49.91 | 75.77 | 78.90 | 81.54 | 87.34 | 61.10 | 64.32 |
| Safe-Net* [72] [TIP'24] | 86.98 | 88.85 | 82.12 | 86.10 | 67.13 | 68.90 | 60.50 | 63.01 | 54.80 | 58.73 | 32.12 | 39.77 | 25.83 | 26.40 | 41.10 | 44.13 | 69.87 | 71.15 | 74.32 | 76.58 | 60.48 | 63.36 |
| CCR* [73] [TCSVT'24] | 92.54 | 93.78 | 85.57 | 87.13 | 67.46 | 68.82 | 55.16 | 59.14 | 63.11 | 60.97 | 27.74 | 31.48 | 23.06 | 46.85 | 51.10 | 54.19 | 75.90 | 79.16 | 81.31 | 87.22 | 62.30 | 66.87 |
| MuSe-Net* [29][PR'24] | 74.48 | 77.83 | 69.47 | 73.24 | 70.55 | 74.14 | 65.72 | 69.70 | 65.59 | 69.64 | 54.69 | 59.24 | 66.64 | 70.55 | 53.85 | 58.49 | 61.05 | 65.51 | 69.45 | 73.22 | 65.15 | 69.16 |
| Ours | 82.78 | 85.18 | 81.46 | 84.03 | 80.34 | 83.11 | 77.60 | 80.67 | 78.75 | 81.69 | 73.38 | 76.94 | 78.41 | 81.40 | 67.22 | 71.06 | 74.20 | 77.63 | 77.26 | 80.27 | 77.14 (+11.99) | 80.20 (+11.04) |
| Satellite → Drone | | | | | | | | | | | | | | | | | | | | | | |
| Zheng et al.[1] [backbone] | 83.45 | 67.94 | 79.60 | 61.12 | 77.60 | 59.73 | 73.18 | 55.07 | 75.89 | 54.45 | 70.76 | 43.26 | 74.75 | 56.44 | 69.47 | 39.25 | 72.18 | 51.91 | 76.46 | 57.59 | 75.33 | 54.68 |
| ResNet-101 [68] [backbone] | 85.73 | 71.79 | 82.45 | 66.46 | 81.46 | 65.68 | 79.74 | 61.72 | 79.74 | 60.59 | 74.75 | 50.31 | 80.17 | 62.61 | 75.32 | 45.37 | 79.60 | 58.21 | 82.31 | 64.67 | 80.13 | 60.74 |
| DenseNet121 [69] [backbone] | 83.74 | 70.34 | 82.31 | 66.32 | 81.17 | 65.23 | 78.60 | 60.33 | 79.46 | 61.66 | 74.61 | 51.14 | 78.46 | 61.68 | 74.47 | 47.88 | 74.32 | 55.26 | 78.32 | 61.63 | 78.55 | 60.15 |
| Swin-T [64] [backbone] | 80.74 | 68.94 | 81.03 | 67.46 | 81.17 | 66.39 | 78.46 | 61.33 | 79.17 | 64.65 | 74.89 | 56.57 | 78.89 | 63.49 | 75.61 | 48.43 | 76.60 | 56.57 | 78.74 | 64.45 | 78.53 | 61.83 |
| IBN-Net [70] [backbone] | 86.31 | 73.54 | 84.59 | 67.61 | 84.74 | 69.03 | 80.88 | 64.44 | 83.31 | 63.71 | 77.89 | 52.14 | 83.02 | 65.74 | 78.46 | 50.77 | 79.46 | 58.64 | 84.02 | 67.94 | 82.27 | 63.36 |
| LPN [2] [TCSVT'21] | 87.02 | 75.19 | 86.16 | 71.34 | 83.88 | 69.49 | 82.88 | 65.39 | 84.59 | 66.28 | 79.60 | 55.19 | 84.17 | 66.26 | 82.88 | 52.05 | 81.03 | 62.24 | 84.14 | 67.35 | 83.64 | 65.08 |
| Sample4Geo* [71] [ICCV'23] | 95.29 | 91.42 | 93.87 | 87.46 | 73.04 | 50.27 | 76.18 | 47.58 | 71.18 | 44.53 | 52.21 | 16.21 | 64.48 | 32.38 | 77.03 | 45.89 | 91.58 | 77.04 | 93.30 | 81.42 | 78.82 | 57.42 |
| Safe-Net* [72] [TIP'24] | 91.22 | 86.06 | 90.04 | 85.43 | 71.12 | 68.56 | 73.26 | 45.62 | 68.23 | 41.78 | 49.32 | 34.72 | 61.07 | 29.86 | 73.15 | 43.08 | 88.54 | 74.65 | 90.02 | 78.21 | 75.69 | 58.80 |
| CCR* [73] [TCSVT'24] | 95.15 | 91.80 | 90.93 | 80.62 | 81.83 | 73.89 | 69.92 | 65.41 | 76.92 | 70.53 | 50.89 | 31.64 | 61.11 | 32.21 | 64.80 | 46.28 | 86.01 | 71.23 | 92.67 | 76.55 | 77.02 | 64.02 |
| MuSe-Net* [29] [PR'24] | 88.02 | 75.10 | 87.87 | 69.85 | 87.73 | 71.12 | 83.74 | 66.52 | 85.02 | 67.78 | 80.88 | 54.26 | 84.88 | 67.75 | 80.74 | 53.01 | 81.60 | 62.09 | 86.31 | 70.03 | 84.68 | 65.75 |
| Ours | 89.16 | 81.80 | 88.73 | 80.58 | 88.16 | 79.87 | 87.59 | 77.25 | 88.45 | 78.20 | 86.73 | 73.23 | 88.59 | 78.14 | 86.59 | 65.20 | 85.31 | 73.25 | 87.88 | 76.33 | 87.72 (+3.04) | 76.39 (+10.64) |

Table 1: **Performance (R@1(%) and AP(%)) on University-1652** for Drone → Satellite and Satellite → Drone tasks. In both tasks, drone-view images are stylized 10 different weather conditions, and the satellite-view images are constant. Best results are highlighted in bold. * denotes the use of official pretrained weights.

output logits $o_b = W_{\mathrm{clf}} z_b + b_{\mathrm{clf}} \in \mathbb{R}^C$, where $W_{\mathrm{clf}} \in \mathbb{R}^{C \times D}$ and $b_{\mathrm{clf}} \in \mathbb{R}^C$. Given the ground-truth location label $y_b \in \{1, \dots, C\}$, the softmax cross-entropy loss is:

$$\mathcal{L}_{\mathrm{CE}} = -\frac{1}{B} \sum_{b=1}^{B} \log \frac{\exp(o_{b,y_b})}{\sum_{c=1}^{C} \exp(o_{b,c})}. \tag{8}$$

**Optimization Objectives.** Our total loss $L_{total}$ is defined as:

$$\mathcal{L}_{total} = \mathcal{L}_{ITC} + \mathcal{L}_{ITM} + \mathcal{L}_{LA} + \mathcal{L}_{CE}. \tag{9}$$

This unified objective enables the model to learn fine-grained, weather-invariant representations, substantially improving cross-weather generalization.

# 4 Experiment

**Implementation Details.** We adopt XVLM [16] as the backbone, which is pre-trained on 4M image–caption pairs, integrates BERT [63] as the text encoder and Swin Transformer [64] as the image encoder. The model is optimized using stochastic gradient descent (SGD) [65] with momentum 0.9 and weight decay 0.0005. Training 210 epochs, with the learning rate reduced by 0.1 at epoch 120 and by 0.01 at epoch 180. We resize input images to $384 \times 384$ pixels and divide them into $32 \times 32$ patches. During training, satellite-view images are augmented via random cropping and horizontal flipping, for drone-view images, we first apply style transformations using the imgaug [61] library and then perform the same random crop and flip augmentations. At test time, we compute the Euclidean distance between query and candidate embeddings to measure similarity. All experiments have been implemented in PyTorch [66] and conducted on a single NVIDIA RTX A6000 GPU, with an average inference time of 0.024s per query.

**Dateset. University-1652** [1] is a large-scale cross-view geo-localization dataset comprising images from 1,652 university locations. Each location is represented by satellite, drone, and ground-level images, with 54 drone-view and 1 satellite-view images per building, as well as street-view imagery. The dataset is split into 701 training and 951 test buildings, with no overlap between train and test sets. **SUES-200** [67] contains multi-view drone and satellite images from 200 locations in Shanghai, encompassing diverse urban scenes, parks, lakes, and public buildings. Drone images are captured from multiple altitudes (150–300m) to simulate varied real-world conditions.

## 4.1 Comparison with Competitive Methods

The experimental results on the University-1652 dataset are shown in Tab 1. We compare the proposed WeatherPrompt with several competitive cross-view geo-localization methods. In the Drone → Satellite task, WeatherPrompt achieves a mean Recall@1 (R@1) accuracy of 77.14% (+11.99%) and a mean Average Precision (AP) of 80.20% (+11.04%), outperforming existing state-of-the-art methods for multi-weather geo-localization. Similarly, in the Satellite → Drone task, WeatherPrompt

| Method | Normal | | Fog | | Rain | | Snow | | Fog+Rain | | Fog+Snow | | Rain+Snow | | Dark | | Over-exp | | Wind | | Mean | |
|---|---|---|---|---|---|---|---|---|---|---|---|---|---|---|---|---|---|---|---|---|---|---|
| | R@1 | AP | R@1 | AP | R@1 | AP | R@1 | AP | R@1 | AP | R@1 | AP | R@1 | AP | R@1 | AP | R@1 | AP | R@1 | AP | R@1 | AP |
| Drone → Satellite | | | | | | | | | | | | | | | | | | | | | | |
| Zheng et al.[1] [backbone] | 57.70 | 58.30 | 48.63 | 49.61 | 53.41 | 52.72 | 41.78 | 43.47 | 37.17 | 37.44 | 44.22 | 46.18 | 40.60 | 40.63 | 23.81 | 25.45 | 49.79 | 50.64 | 47.42 | 48.31 | 44.43 | 45.12 |
| IBN-Net [70] [backbone] | 65.34 | 63.78 | 56.03 | 56.57 | 55.73 | 58.55 | 47.80 | 49.53 | 43.45 | 44.98 | 50.04 | 51.00 | 45.51 | 45.92 | 29.61 | 30.93 | 56.01 | 56.96 | 57.36 | 58.10 | 50.69 | 51.63 |
| Sample4Geo† [71] [ICCV'23] | 74.93 | **78.76** | 72.58 | **76.44** | 34.60 | 41.56 | 28.95 | 35.02 | 35.10 | 41.47 | 12.95 | 17.90 | 20.05 | 25.95 | 34.18 | 38.99 | 38.40 | 43.68 | **67.80** | **72.41** | 41.95 | 47.22 |
| Safe-Net* [72] [TIP'24] | 76.31 | 75.35 | **73.53** | 73.44 | 54.15 | 55.05 | 48.94 | 50.10 | 45.12 | 47.92 | 40.05 | 40.18 | 25.95 | 26.12 | 29.74 | 31.48 | 54.86 | 58.68 | 58.10 | 58.95 | 50.68 | 51.63 |
| CCR† [73] [TCSVT'24] | 73.22 | 74.53 | 70.95 | 73.14 | 60.14 | 64.95 | 50.31 | 53.12 | 45.87 | 49.14 | 45.80 | 47.87 | 31.25 | 32.94 | 31.03 | 34.36 | 59.97 | 61.07 | 52.02 | 53.33 | 52.06 | 53.46 |
| MuSe-Net† [29] [PR'24] | 66.07 | 67.02 | 58.49 | 59.65 | 58.94 | 60.14 | 54.85 | 56.12 | 44.31 | 45.82 | 49.81 | 51.26 | 49.42 | 50.87 | 29.34 | 31.03 | 55.02 | 56.36 | 59.97 | 61.05 | 52.02 | 53.33 |
| Ours | **76.72** | 75.51 | 68.49 | 68.87 | **71.77** | **71.20** | **59.95** | **60.62** | **58.24** | **58.83** | **64.36** | **66.27** | **58.49** | **58.89** | **40.42** | **55.75** | **61.57** | **71.70** | 65.19 | 67.00 | **62.52** (+10.46) | **63.26** (+9.80) |
| Satellite → Drone | | | | | | | | | | | | | | | | | | | | | | |
| Zheng et al.[1] [backbone] | 70.20 | 57.98 | 63.77 | 46.90 | 68.72 | 50.85 | 61.72 | 39.70 | 62.10 | 32.75 | 71.70 | 40.39 | 59.72 | 37.55 | 45.49 | 25.28 | 52.11 | 43.40 | 56.62 | 45.31 | 61.21 | 42.01 |
| IBN-Net [70] [backbone] | 73.68 | 62.91 | 67.41 | 55.75 | 72.30 | 56.44 | 64.07 | 47.69 | 66.98 | 39.54 | 71.10 | 47.32 | 68.46 | 45.95 | 54.72 | 31.53 | 65.64 | 53.77 | 73.48 | 57.03 | 67.79 | 49.79 |
| Sample4Geo† [71] [ICCV'23] | 87.50 | 79.57 | 83.75 | 71.14 | 42.50 | 25.24 | 40.00 | 21.59 | 38.75 | 23.22 | 30.00 | 10.58 | 26.25 | 16.44 | 56.25 | 29.75 | 58.75 | 30.38 | **83.75** | **69.66** | 54.75 | 37.76 |
| Safe-Net* [72] [TIP'24] | 88.31 | 80.35 | 81.33 | 68.60 | 40.21 | 41.04 | 36.43 | 37.50 | 33.12 | 35.45 | 24.78 | 27.65 | 41.12 | 32.31 | 53.88 | 27.01 | 54.19 | 57.82 | 79.36 | 57.09 | 53.27 | 46.48 |
| CCR† [73] [TCSVT'24] | 90.59 | 80.45 | 82.99 | 70.62 | 43.39 | 45.90 | 39.81 | 40.88 | 42.63 | 39.46 | 29.32 | 30.65 | 25.89 | 26.94 | 26.01 | 30.40 | 58.01 | 59.13 | 83.09 | 61.05 | 52.17 | 48.55 |
| MuSe-Net† [29] [PR'24] | 76.56 | 66.02 | 72.19 | 57.87 | 72.19 | 58.11 | 68.38 | 51.22 | 66.56 | 42.25 | 69.06 | 46.80 | 69.38 | 47.79 | 53.75 | 27.94 | 70.00 | 52.67 | 76.25 | 60.74 | 69.43 | 51.14 |
| Ours | **90.61** | **81.24** | **86.14** | **71.15** | **83.94** | **73.80** | **71.03** | **60.19** | **84.41** | **58.49** | **79.16** | **64.93** | **77.28** | **60.15** | **56.75** | **47.85** | **81.65** | **74.04** | 80.30 | 69.38 | **80.73** (+11.30) | **66.12** (+14.98) |

Table 2: **Performance (R@1(%) and AP(%)) on SUES-200** for Drone → Satellite and Satellite → Drone tasks. In both tasks, drone-view images are stylized in 10 weather conditions, while satellite-view images remain constant. Best results are highlighted in bold. * denotes the use of official pretrained weights. † denotes the use of official pretrained weights on University-1652.

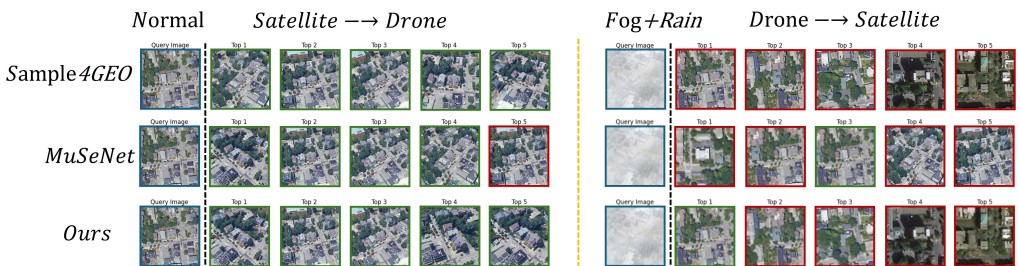

Figure 4: **Qualitative comparison under varying weather.** While existing methods perform reliably under clear weather, their accuracy drops markedly in adverse conditions. Our approach maintains superior localization performance, especially when drone images are severely affected by weather. Green boxes indicate correct matches, while images in red boxes represent incorrect matches.

attains a mean R@1 accuracy of 87.72% (+3.04%) and a mean AP of 76.39% (+10.64%). Our method validates competitive performance when handling drone view images under different conditions, especially in terms of cross-view geo-localization in challenging weather scenarios (*e.g.*, Dark, Fog+Snow, and Rain+Snow). In addition to quantitative comparisons, we present qualitative retrieval results in Figure 4. While existing methods can retrieve plausible matches in clear conditions, they suffer a substantial performance drop when encountering adverse weather conditions. In contrast, our approach consistently retrieves more accurate matches among the top-ranked candidates even in the presence of severe weather, highlighting its robustness and generalization capability. In the real-world captured SUES-200 dataset, we combine drone-view images captured at four altitudes (150m, 200m, 250m, and 300m) to simulate diverse operational heights in practical drone applications, enabling a comprehensive evaluation of our method. As shown in Tab 2, compared to the state-of-the-art multi-weather geolocalization method MuSe-Net, our method shows significant improvements in several metrics. Specifically, Ours improves mean R@1 accuracy in the Drone → Satellite task from 52.02% to 61.20% (+9.18%) and mean AP from 53.33% to 63.26% (+9.80%). In the Satellite → Drone task, our method increases mean R@1 accuracy from 69.43% to 80.73% (+11.30%) and mean AP from 51.14% to 66.12% (+14.98%).

### 4.2 Ablation Studies and Further Discussion

**Impact of Weather-Driven Gating Mechanism.** We conduct ablation experiments to quantify the impact of the weather-driven gating module on multimodal fusion. Table 3a summarizes three alternative fusion strategies under identical settings: (1) Concatenation, which concatenates visual and textual features directly; (2) Static Gate, which fuses modalities using fixed average weights; and (3) Dynamic Gate (Ours), which adaptively reweights visual channels based on weather semantics. We find that the dynamic gating mechanism improves mean AP by 2.3% over the no-gate baseline, with only 0.2M extra parameters and 0.4 ms added latency. This validates that weather-aware, channel-wise fusion enables more reliable feature integration, especially under adverse weather. The dynamic gate also reduces degradation from ambiguous weather descriptions, demonstrating its robustness for cross-weather geo-localization.

**Impact of Text-Guided Weather Semantics.** We conduct an ablation study to assess how multi-weather textual supervision affects cross-view geo-localization. Table 3b reports mean Recall@1

| Method | D2S | | | | S2D | | | |
| | Normal | | Mean | | Normal | | Mean | |
| | R@1(%) | AP(%) | R@1(%) | AP(%) | R@1(%) | AP(%) | R@1(%) | AP(%) |
|---|---|---|---|---|---|---|---|---|
| +Concatenation | 81.59 | 84.11 | 75.73 | 78.90 | 88.59 | 81.76 | 85.38 | 73.84 |
| +Static Gate | 82.35 | 84.75 | 75.26 | 78.45 | 88.45 | **82.25** | 85.89 | 74.61 |
| +Dynamic Gate | **82.78** | **85.18** | **77.14** | **80.20** | **89.16** | 81.80 | **87.72** | **76.39** |

(a) Impact of Weather-Driven Channel Gating on University-1652

| CoT Step | D2S | | S2D | |
| | Mean R@1(%) | Mean AP(%) | Mean R@1(%) | Mean AP(%) |
|---|---|---|---|---|
| NAN | 74.53 | 78.72 | 85.17 | 73.26 |
| 0 | 75.10 | 79.15 | 85.45 | 73.65 |
| 2 | 75.35 | 79.10 | 85.85 | 74.40 |
| 4 | 76.05 | 79.80 | 86.60 | 75.20 |
| 6 | **77.14** | **80.20** | **87.72** | **76.39** |

(b) Different CoT Setting on University-1652

| | | Dark+Rain+Fog | | | |
| | | R@1(%) | R@5(%) | R@10(%) | AP(%) |
|---|---|---|---|---|---|
| D2S | Sample4Geo [71] | 22.22 | 61.11 | 77.78 | 31.65 |
| | MuSe-Net [29] | 22.22 | 66.67 | 83.33 | 31.70 |
| | Ours | **44.44** | **83.33** | **94.44** | **64.94** |
| S2D | Sample4Geo [71] | 33.33 | 61.11 | 83.33 | 43.47 |
| | MuSe-Net [29] | 38.89 | 61.11 | 88.89 | 44.34 |
| | Ours | **66.66** | **77.78** | **94.44** | **72.15** |

(c) Evaluation on Real World Videos

Table 3: **Ablation Studies on University-1652 and Real-World Videos.** D2S denotes the Drone $\rightarrow$ Satellite and S2D denotes the Satellite $\rightarrow$ Drone.

and mean Average Precision (AP) across two tasks (Drone $\rightarrow$ Satellite and Satellite $\rightarrow$ Drone) on University-1652, under varying levels of Chain-of-Thought (CoT) guidance. The no-caption baseline (NAN) removes all text-based components, training the model with only the visual encoder and classification head. When progressively increasing the CoT prompt steps, we observe consistent performance gains for both tasks. Compared to the baseline, the best CoT setting (6-step) achieves improvements of +2.61% mean Recall@1 and +1.48% mean AP in the Drone $\rightarrow$ Satellite task, and +2.55% mean Recall@1 and +3.13% mean AP in the Satellite $\rightarrow$ Drone task. Notably, all models leveraging text-based weather guidance achieve higher mean Recall@1 and AP compared to the visual-only baseline, demonstrating that even minimal textual supervision provides substantial benefits for model robustness and accuracy.

**Impact of Prompt Structuring on Multimodal Alignment.** To comprehensively assess the impact of stepwise reasoning in prompt engineering on multimodal alignment, we generate captions using the same large vision–language model [17], varying only the number of reasoning steps prescribed in the prompt. The zero-step baseline produces a single-turn weather description without explicit intermediate reasoning. In contrast, the Chain-of-Thought (CoT) variants employ multi-stage reasoning prompts, including 2-step, 4-step, and 6-step configurations. The optimal 6-step CoT scheme decomposes the generation process into two reasoning chains: the first three steps sequentially estimate global visibility, local atmospheric cues, and fine-grained weather semantics; the latter three steps, conditioned on the inferred weather prior, successively infer macro scene layout, enumerate structural elements, and capture fine-grained spatial relationships. This structured decomposition compels the model to capture both meteorological variations and spatial semantics, yielding captions that are both semantically rich and consistent in format. As shown in Table 3b, model performance steadily improves as the number of reasoning steps increases, with the 6-step CoT achieving the best results in terms of both Recall@1 and mAP. These findings show that fine-grained, multi-stage reasoning significantly enhances cross-weather robustness and discriminative capacity, serving as a crucial ingredient for high-quality vision–language alignment in challenging multi-weather scenarios.

**Real-World Performance under Adverse Weather Conditions.** To validate real-world robustness, we collected 54 drone-satellite video pairs from YouTube to evaluate ours under three weather conditions, including Dark, Rain, and Fog. As shown in Tab 3c, our model consistently achieves superior results in terms of R@1, R@5, R@10, and Average Precision (AP). These results underscore the strong robustness and generalization capability of our model under challenging real-world scenarios with poor lighting or inclement weather conditions.

**Limitations.** Despite of significant advancements in cross-view geo-localization under diverse weather conditions, several limitations inherited from external components warrant discussion: (1) Dataset. The evaluation relies on existing datasets, which lack exhaustive geographic and weather diversity. Their limited scope in representing globally rare or region-specific weather phenomena may affect generalization to unseen environmental extremes. (2) Language Model Biases. The weather and spatial captions generated by off-the-shelf vision-language models inherit biases from

their pretraining corpora. Subtle inaccuracies in descriptor granularity, *e.g.*, "haze" vs. "fog", could propagate into the alignment process. Dataset and LVLM advances will mitigate these limitations.

## 5 Conclusion

In this paper, we propose a novel training-free, text-guided multi-modality alignment framework for robust cross-view geo-localization under complex and unseen weather conditions. By leveraging large vision–language models, we introduce a reasoning pipeline that automatically generates high-fidelity weather and spatial captions via chain-of-thought prompting, eliminating the need for costly manual descriptions or expert-controlled. Our multi-modal alignment model incorporates a dynamic channel-wise gating mechanism that adaptively fuses textual weather semantics with visual representations, achieving fine-grained disentanglement of scene and weather features. Extensive experiments on the University-1652 and SUES-200 benchmarks validate that our method consistently outperforms state-of-the-art approaches, particularly in challenging multi-weather scenarios. Our approach sets a new paradigm for leveraging language-driven priors in aerial geo-localization and offers a scalable path toward real-world deployment under diverse environmental conditions.

## 6 Acknowledgement

This work is supported by the Shanghai Committee of Science and Technology, China (Grant No.23ZR1423500), the National Natural Science Foundation of China under Grant No.62302287, University of Macau MYRG-GRG2024-00077-FST-UMDF and SRG2024-00002-FST, and the Science and Technology Development Fund (FDCT) 0043/2025/RIA1.

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
