# OpenReview forum: "WeatherPrompt: Multi-modality Representation Learning for All-Weather Drone Visual Geo-Localization"
_NeurIPS.cc/2025/Conference — NeurIPS 2025 poster_

### Official Review · Reviewer_BCVv · 2025-06-19

**Clarity:** 3
**Significance:** 3
**Originality:** 3
**Rating:** 5
**Confidence:** 4

**Summary:**

This paper proposes WeatherPrompt, a novel multi-modal learning framework for robust drone visual geo-localization under diverse weather conditions. The key innovation lies in leveraging CoT prompting with LVLMs to generate structured weather-aware textual descriptions without manual labeling, enabling scalable training data creation. The framework further introduces a text-driven dynamic gating mechanism to disentangle scene and weather features adaptively, enhanced by multi-modal contrastive and matching losses. Experiments on University-1652 and SUES-200 datasets demonstrate significant improvements over state-of-the-art methods, particularly in challenging conditions (e.g., +18.69% Recall@1 for fog/snow).

**Questions:**

(1) The article points out that large vision-language models usually generate weather labels or descriptions in an intuitive manner. However, how can you ensure that the descriptions guided by CoT prompts are free from factual biases?
The current approach only improves the accuracy of the descriptions provided by LLMs, but it cannot supervise whether these descriptions are accurate.

(2) Expansion of (1): The 6-step CoT prompt is central to description quality. Was this structure empirically optimized? A comparison with simpler prompts (e.g., direct QA) would help justify the complexity.

(3)Regarding the second challenge in the intro: (2) Scene–Weather Feature Entanglement is somewhat ambiguous.
The entanglement of scene and weather features here is not clearly explained. Both the text description and the image features should contain scene and weather information. The disentanglement here should be targeted at the scene and weather in the two modalities.

(4) You report strong performance on Dark+Rain+Fog, but the dataset’s synthetic weather may not capture real-world complexity. How does the model handle natural weather transitions without explicit CoT supervision during inference?

**Ethical Concerns:**

["NO or VERY MINOR ethics concerns only"]

**Final Justification:**

The author has solved all my problems, thus I have decided to raise my rating.

**Limitations:**

Yes.

**Paper Formatting Concerns:**

There is no.

**Quality:**

3

**Strengths And Weaknesses:**

Strengths:

1. The method is intuitive. Currently, it is a trend to use large language models (LLMs) for scene description. However, direct image description by LLMs may lead to hallucination, while using chain-of-thought (CoT) can guide the output of LLMs to meet expectations.

2. The training-free description pipeline generalizes to unseen or mixed weather conditions, addressing a key limitation of closed-set methods. Currently, many methods utilize learning from synthetic data to generalize to the real world.

3. Comprehensive benchmarks across 10+ weather types and ablation studies validate the design choices

Weaknesses:

1. The method is intuitive but overly simplistic. Regarding the proposed scene description method, although it is training-free, its corresponding contribution points are much fewer. Moreover, the multimodal alignment framework also lacks key innovative points.

2. I still have some questions to ask you, all of which are included in Questions.

---

> ### Author Rebuttal · Authors · 2025-07-30
>
> **Q1**: The method intuitive yet overly simplistic, with limited contributions from the training‑free scene description and insufficient innovation in the multimodal alignment framework.
>
> **A1**: We thank the reviewer for the feedback. **Although our method is simple, it is effective.** It achieves the highest combined mean R@1 and AP across all weather conditions, significantly outperforming existing baselines and demonstrating its efficacy (see Tables 1 and 2 in paper). **Our main contribution is to address the unresolved challenge of generalizing to unseen/mixed weather, proposing the new “train once, inference without finetuning” all‑weather cross‑view geolocalization framework.**
>
> **Previous coarse-grained pseudo-weather labels (e.g., simply “Rain” or “Fog”) provide only category-level information and cannot describe the intensity or duration of a weather event.** By leveraging fine‑grained description along with image‑text contrastive (ITC) and matching (ITM) losses, we achieve efficient multimodal alignment that balances simplicity with practical deployment needs.
>
> **Q2**: How to ensure that CoT prompt guided weather descriptions are factually unbiased and accurate, given that the current approach cannot verify their correctness.
>
> **A2**: **We randomly sampled instances and audited the factuality of the CoT weather texts using an independent LLM judge (Llama) and three cs PhD. The accuracy of the weather terms is about 70%, with high agreement between human and model scores.** Most errors arise from boundary/compound states (e.g., “Fog” or “Fog+Snow/Rain”), where continuous intensities and mixtures are hard to express with discrete words. We use a fixed‑template chain‑of‑thought prompt that outputs only objective physical attributes (brightness, visibility, shadows, and major structural orientations), minimizing the introduction of subjective or biased tokens. Because these descriptions serve solely for channel‑level feature re‑weighting, any occasional biased or irrelevant tokens see their weights driven toward zero under our joint contrastive and classification supervision. To demonstrate our method’s independence from a specific LLM, we replaced the caption generator with “GLM‑4.1V‑9B‑Thinking”, retrained and retested, R@1 and mAP remained consistent (see Table A). We also validated cross‑weather robustness on existing web‑collected real‑world multi‑weather datasets (see Table 3(c), Page 8). We agree on the need for larger‑scale, complex real‑weather evaluations. There is no publicly available real‑world multi‑weather drone dataset, collecting real multi‑weather data is costly and time‑consuming. So we are actively collecting a multi‑weather drone dataset for future release.
>
> | **Method**           | **D2S  mR@1 (%)**  | **D2S mAP (%)**   | **S2D mR@1 (%)**  | **S2D mAP (%)**   |
> |:---------------------|:---------------:|:---------------:|:---------------:|:---------------:|
> | **GLM**  |  76.52  |  79.85  |  86.62  |  76.33 |
> | **Ours**             |      77.14      |      80.20      |      87.72      |      76.39      |
>
> *Table A: Difference Description Generator Results on the University‑1652 Dataset. (D2S: drone‑to‑satellite, S2D: satellite‑to‑drone, mR@1: mean R@1, mAP: mean AP, GLM: description generator with “GLM‑4.1V‑9B‑Thinking”, Ours: our paper results.)*
>
> **Q3**: Expansion of (1): The 6-step CoT prompt is central to description quality. Was this structure empirically optimized? A comparison with simpler prompts (e.g., direct QA) would help justify the complexity.
>
> **A3**: **The structure has been empirically optimized.** We evaluated the 6‑step CoT design (L276–L287, Table 3b, Page8): a single‑sentence prompt yields only modest gains over the Image‑only variant, and simplified CoT versions (2/4 steps) underperform the 6‑step design.
>
> **Q4**: The description of scene–weather feature entanglement ambiguous and asks how disentanglement specifically separates scene and weather information across both image and text modalities.
>
> **A4**: We thank the reviewer for this insightful question. Unlike prior works that inject coarse tags like rain or fog directly, we assign each image a fine-grained sentence-level weather description. We apply the image‑text contrastive loss (ITC) and the image‑text matching loss (ITM) to guide the network in capturing subtle weather variations within its feature representations. For example, conventional methods tag all precipitation as rain treating drizzle and heavy downpour as the same class. Our chain of thought prompt generates detailed weather descriptions such as drizzle showers heavy rain or hail so that the network can adjust feature weights based on specific weather information instead of relying on a single rain tag resulting in more accurate representation of weather cues.
>
> **Q5**: How the model manages natural weather transitions during inference without explicit CoT supervision given that synthetic weather may not capture real world complexity.
>
> **A5**: The synthetic “Dark + Rain + Fog” augmentation is not produced by overlaying three fixed labels. Instead, **imgaug continuously samples augmentation parameters at random, generating a vast number of different combinations during training effectively scattering points throughout the parameter space.** Real video phenomena such as gradual darkening, intensifying fog, and variable rainfall correspond to specific values or transitions between values in this space. Because the sampling is dense, the states of real frames are “covered” by adjacent training samples, enabling the model to generalize to natural intermediate weather conditions. At inference, we modulate features using pre‑generated CoT descriptions that encode both weather and spatial information. As a result, when weather changes slowly or transitions smoothly in natural environments, our model maintains high robustness without requiring frequent regeneration of descriptions. **Moreover, we evaluated three weather conditions unseen during training. In both drone to satellite and satellite to drone retrieval tasks, our method consistently outperforms existing approaches in R@1 and AP (Table B).**
>
> | **Method**   | **Task** | **Winter Night R@1 (%)** | **Winter Night AP (%)** | **Freezing Rain R@1 (%)** | **Freezing Rain AP (%)** | **Blizzard R@1 (%)** | **Blizzard AP (%)** |
> |:-------------|:--------:|:-----------------------:|:----------------------:|:-------------------------:|:------------------------:|:--------------------:|:-------------------:|
> | CCR          | S2D      |  69.97                  |  44.08                 |  66.16                    |  55.20                   |  60.88              |  34.23             |
> | SafeNet      | S2D      |  56.92                  |   8.28                 |  41.08                    |   8.15                   |  57.92              |  22.05             |
> | Sample4Geo   | S2D      |  58.35                  |  21.33                 |  65.62                    |  32.76                   |  66.90              |  32.47             |
> | MuseNet      | S2D      |  73.32                  |  40.19                 |  76.03                    |  43.51                   |  69.76              |  35.21             |
> | **Ours**     | S2D      | **82.60**               | **59.43**              | **85.31**                 | **61.17**                | **71.90**           | **41.94**          |
> | CCR          | D2S      |  36.05                  |  38.79                 |  47.29                    |  51.70                   |  25.73              |  30.07             |
> | SafeNet      | D2S      |   3.25                  |   4.87                 |   2.64                    |   4.30                   |  11.46              |  15.43             |
> | Sample4Geo   | D2S      |  25.40                  |  28.60                 |  37.83                    |  41.95                   |  36.00              |  40.42             |
> | MuseNet      | D2S      |  39.45                  |  44.36                 |  44.09                    |  49.01                   |  33.44              |  38.51             |
> | **Ours**     | D2S      | **58.67**               | **63.08**              | **61.52**                 | **65.78**                | **59.76**           | **64.15**          |
>
> *Table B: Results on the University‑1652 dataset under three unseen weather conditions.*

---

> ### Comment · Reviewer_BCVv · 2025-08-03
> **Official Comment by Reviewer BCVv**
>
> Appreciate the detailed rebuttal.
> The author has solved all my problems, and I have decided to raise my rating.

---

> > ### Author Response · Authors · 2025-08-03
> >
> > We sincerely appreciate your thorough and thoughtful review, and are truly grateful for your recognition and positive feedback on our work.

---

### Official Review · Reviewer_Favf · 2025-06-22

**Clarity:** 2
**Significance:** 3
**Originality:** 3
**Rating:** 3
**Confidence:** 4

**Summary:**

WeatherPrompt introduces a training-free, text-guided multimodal framework for drone geo-localization under diverse weather conditions by leveraging large vision–language models and chain-of-thought prompting to generate structured weather and spatial descriptions. Through a dynamic gating mechanism and contrastive alignment between text and image features, the model achieves robust performance across unseen and adverse weather scenarios, outperforming prior state-of-the-art methods.

**Questions:**

Please refer to the weaknesses section.

**Ethical Concerns:**

["NO or VERY MINOR ethics concerns only"]

**Final Justification:**

Most of my concerns have been addressed. Therefore, I am willing to raise my score to borderline reject. Other reviewers have given positive feedback, and I hope the author can review the entire text again, as some minor errors were found during the review process that significantly affect the reading experience.

**Limitations:**

Yes

**Paper Formatting Concerns:**

No formatting concerns.

**Quality:**

2

**Strengths And Weaknesses:**

Strengths:
1. The issue studied in the paper is interesting. It is largely overlooked by existing methods.
2. The abs and intro sections are well-structured and easy to follow.

Weaknesses:
1. For Figure 3, how to get the box ( $I$, $I_1$, ... )?
2. Line 178, the authors claim that their method is training-free, but why is it training-free? There are learnable parameters in the pipeline.
3. Line 181, it's confused between Figure 3 and the illustration here. The cross-modal fusion module is not reflected in Figure 3.
4. Lines 185-198 are not clearly written. It is ambiguous whether $T$ refers to the different colored segments in Figure 3 or the entire weather descriptions. Simply referring to "text embeddings" is too vague.
5. For line 193, the samples, What is $C$?
6. Line 211, MPL should be MLP.
7. In Table 1, the normal part, the performance drop seems too large. Would training Sample4Geo on the augmented data directly lead to good localization results under different weather conditions?
8. A very important set of experiments is missing: Why are these four loss functions necessary? Do all of them actually contribute positively? Do they have the same impact under different weather conditions?
9. Additionally, there is no analysis of hyperparameters. Do the weights of these losses have to be equal?
10. Why this paper is under "Reinforcement learning" primary area?
11. The code is not open released.

---

> ### Author Rebuttal · Authors · 2025-07-30
>
> **Q1**: For Figure 3, how to get the box？
>
> **A1**: **As indicated in the caption of Fig. 1 (Page 2): "We leverage Off the-shelf Visual Grounding Model (VGM), i.e., XVLM."** Its grounding branch outputs candidate boxes and confidences for each phrase, we take the highest‑confidence box per phrase to draw Fig. 3, with no additional training.
>
> **Q2**: Line 178, the authors claim that their method is training-free, but why is it training-free? There are learnable parameters in the pipeline.
>
> **A2**: "Training-free" indicates that the proposed model can handle unseen weather conditions without requiring additional fine-tuning.
> In contrast to existing methods, which typically necessitate fine-tuning for new weather scenarios, our approach leverages text description inputs to achieve immediate adaptation to novel environments.
>
> **We clarify this around lines 180–187 on Page 5**: the classification head only provides supervision during training and is discarded at inference. **Pages 2 (Lines 60–62) describe the inference procedure: "avoiding any additional online
> fine-tuning or dedicated parameter sets for different weather conditions."**
> At inference time we simply compute L2‑normalized similarities between query and gallery embeddings, no further training or fine‑tuning is performed, yielding a training‑free pipeline.
>
> **Q3**: Line 181, it's confused between Figure 3 and the illustration here. The cross-modal fusion module is not reflected in Figure 3.
>
> **A3**: **The cross-modal fusion is implemented by the right-hand gray block in Fig. 3 (Self-Attention / Cross-Attention / Feed-Forward).** As is standard practice in cross-modal architectures, image tokens serve as queries while the weather text and spatial phrase tokens provide keys/values. It takes the weather text embedding (top), spatial phrase embeddings (bottom), and image features (center), via cross‑attention it injects textual semantics into the image channels and outputs the fused features used by the ITM / LA / CE losses.
>
> **Q4**: Lines 185-198 are not clearly written. Simply referring to "text embeddings" is too vague.
>
> **A4**: “Text embedding” comprises two types: (1) Weather text embedding: the full weather description (top of Fig.3) is fed to the text encoder and the [CLS] token is taken as the global vector $f_t$. (2) Phrase level spatial text embeddings: the lower paragraph is split by color into three phrases (corresponding to regions $I_1/I_2/I_3$ in the figure). Each phrase is fed to the same text encoder to obtain $f_{t1}, f_{t2}, f_{t3}$ (marked with matching colors in Fig.3). In the revision, we will rename “text embedding” to “weather text embedding” and “phrase level spatial text embeddings,” and add corresponding labels in Fig.3 to remove ambiguity.
>
> **Q5**: For line 193, the samples, What is C？
>
> **A5**: C was intended to denote the image description, the correct symbol in the paper should be T. Thank you for pointing out the mistake.
>
> **Q6**: Line 211, MPL should be MLP？
>
> **A6**: Yes, it is MLP. We’ll correct it in the revision. Thank you for pointing out the mistake.
>
> **Q7**: Would training Sample4Geo on the augmented data directly lead to good localization results under different weather conditions?
>
> **A7**: We thank the reviewer for raising this important question. **Our setting is training free, we do not perform any retraining or finetuning at inference, so the initial baselines were also evaluated without any weather‑specific retraining.** We then added results on University‑1652 dataset where methods are trained with the same multi‑weather random augmentations as ours. To address the reviewer’s concern, **we then fine-tuning three baseline models (Sample4Geo, Safe‑Net, and CRR) using the exact same multi‑weather random augmentations as ours and report their performance in Table A.** All three baselines exhibit average improvements over their original scores, yet our method still outperforms them by a large margin. On drone‑to‑satellite retrieval, the R@1 and AP gaps remain above 10%; on satellite‑to‑drone retrieval, the gaps exceed 6% and 15%, respectively. These results confirm that our robustness gains are not merely due to exposure to additional augmentations.
>
> | **Method**           | **D2S  mR@1 (%)**  | **D2S mAP (%)**   | **S2D mR@1 (%)**  | **S2D mAP (%)**   |
> |:---------------------|:---------------:|:---------------:|:---------------:|:---------------:|
> | **Sample4Geo (FT)**  |  66.72 (+5.62)  |  70.10 (+5.78)  |  81.63 (+2.81)  |  62.29 (+4.87)  |
> | **Safe‑Net (FT)**    |  62.47 (+1.99)  |  65.99 (+2.63)  |  77.18 (+1.49)  |  59.20 (+0.40)  |
> | **CRR (FT)**         |  68.68 (+6.38)  |  72.09 (+5.22)  |  86.55 (+9.53)  |  66.02 (+2.00)  |
> | **Ours**             |      77.14      |      80.20      |      87.72      |      76.39      |
>
> *TableA: Multi‑weather Fine‑Tuning Results on the University‑1652 Dataset. (D2S: drone‑to‑satellite, S2D: satellite‑to‑drone, mR@1: mean R@1, mAP: mean AP, FT: fine‑tuning, Ours: our paper results. The brackets indicate the results of the increase on the original method.)*
>
> **Q8**: Why are these four loss functions necessary? Do all of them actually contribute positively? Do they have the same impact under different weather conditions?
>
> **A8**: **We follow common practice in vision language retrieval and set the weights of ITC and ITM to 1 : 1**, prior work such as ALBEF (Li et al., 2021), BLIP (Li et al., 2022), FLAVA (Singh et al., 2022), SIMLA (Khan et al., 2022), XVLM (Zeng et al., 2021) and more have also adopted equal weighting of the image‑text contrastive and matching losses, demonstrating stable performance.
>
> **Following prior work, we keep ITC and ITM as our standard objectives for cross modal retrieval and ablate the other two losses**, LA and CE. Table B shows that both losses contribute positively: CE provides a clearer global margin, while LA yields larger gains under low visibility conditions (Dark and Fog) and only minor changes under Normal. In summary, ITC and ITM enable cross modal alignment, CE enforces inter class discrimination, and LA stabilizes local semantic alignment, and they form a complementary set of objectives.
>
> | **Method**           | **D2S  mR@1 (%)**  | **D2S mAP (%)**   | **S2D mR@1 (%)**  | **S2D mAP (%)**   |
> |:---------------------|:---------------:|:---------------:|:---------------:|:---------------:|
> | **W/O LA**  |  76.57  |  79.70  |  86.58  |  75.53  |
> | **W/O CE**    |  75.31  |  78.55  |  86.44  |  75.40  |
> | **Ours**             |      77.14      |      80.20      |      87.72      |      76.39      |
>
> *Table B: Ablation Study of Loss Necessary Results on the University‑1652 Dataset. (W/O LA: without LA loss, W/O CE: without CE loss.)*
>
> **Q9**: Do the weights of these losses have to be equal?
>
> **A9**: **We follow common practice in vision and language retrieval and set the weights of ITC and ITM to 1:1; prior work such as CLIP, ALIGN and more, also adopts equal weighting with stable performance.** Localized Alignment (LA) loss acts as a training time localization consistency regularize and is included in the total loss to stabilize phrase–region alignment and shape the embedding space. **At inference, the retrieval score is computed only from the similarity between L2 normalized image and text embeddings, the scoring function does not include LA.** In implementation, we normalize the box error and average over boxes so that its gradient magnitude is comparable to ITC and ITM, hence no separate reweighting is required.
>
> To validate our weighting scheme, **we conducted two ablation studies: (1) 2·(ITC+ITM) : 1·LA : 1·CE and (2) 1·(ITC+ITM) : 1·LA : 2·CE.** As shown in Table C, the equal weight configuration (Ours) is best on both D2S and S2D; moderately increasing the contrastive term causes small fluctuations, while increasing CE significantly reduces R@1 and AP. Specifically, relative to Ours, 2·(ITC+ITM) lowers D2S R@1 and AP by 1.22% and 0.95%, and lowers S2D R@1 and AP by 0.43% and 0.33%. In 2·CE settings lowers D2S R@1 and AP by 5.84% and 5.91%, and lowers S2D R@1 and AP by 3.57% and 4.64%. These results indicate that equal ITC and ITM with balanced LA and CE is a robust choice.
>
> | **Method**           | **D2S  mR@1 (%)**  | **D2S mAP (%)**   | **S2D mR@1 (%)**  | **S2D mAP (%)**   |
> |:---------------------|:---------------:|:---------------:|:---------------:|:---------------:|
> | **2(ITC+ITM)**  |  75.92  |  79.25  |  87.29  |  76.06  |
> | **2CE**    |  71.30  |  74.29  |  84.15  |  71.75  |
> | **Ours**             |      77.14      |      80.20      |      87.72      |      76.39      |
>
> *Table C: Difference Loss Weighting Scheme Results on the University‑1652 Dataset.*
>
> **Q10**: Why this paper is under "Reinforcement learning" primary area?
>
> **A10**: At submission time, the system did not offer a “Representation Learning” option. Given that our task is drone (robotics) cross‑view geolocalization and the learned representation directly serves as a precursor state estimate for decision and control (RL/planning), we selected Reinforcement Learning (decision & control, robotics) as the closest area.
>
> **Q11**: The code is not open released.
>
> **A11**: We will open source the entire code shortly, including trained weights and the collected real world dataset.

---

> > ### Comment · Reviewer_Favf · 2025-08-01
> >
> > Thank you for the response. It has addressed my concerns, and I will raise my score.

---

> > > ### Author Response · Authors · 2025-08-03
> > >
> > > We sincerely appreciate your positive feedback and are pleased that our response has resolved your concerns. Your willingness to raise the score is deeply appreciated and holds great significance for us.

---

### Official Review · Reviewer_ZGxr · 2025-07-01

**Clarity:** 4
**Significance:** 3
**Originality:** 3
**Rating:** 4
**Confidence:** 4

**Summary:**

This paper proposes a multimodal learning framework named WeatherPrompt, which aims to address the performance degradation of drone-based visual geo-localization under complex weather conditions. This framework establishes weather-invariant representations by fusing image embeddings with textual context to enhance generalization capability across diverse weather scenarios.
The main innovations include: 1) A training-free weather reasoning mechanism that automatically generates high-quality weather descriptions using large multimodal models; 2) A text-driven multimodal architecture that disentangles scene and weather features through dynamic gating mechanisms; 3) Comprehensive validation demonstrating superior performance across multiple benchmark datasets.

**Questions:**

1. On p.2, line 74: Add “@” between “Recall” and “1” (typo correction).
2. How do the authors plan to further validate the generalization capability of WeatherPrompt on real-world datasets with more diverse and complex weather conditions? Are there any ongoing efforts to collect and test on additional real-world drone geo-localization datasets? Related rating: significance.
3. Given the use of large pre-trained models like Qwen2.5-VL-32B, what are the computational requirements and potential bottlenecks for deploying WeatherPrompt in real-time applications, especially on resource-constrained platforms? It is important to note that the inference time reported in the paper (0.024 seconds per query) does not include the time required to generate the text descriptions, which is a necessary step for each inference in practical applications. How does this additional time impact the feasibility of real-time deployment, and what strategies can be employed to mitigate this potential bottleneck? Related rating: quality.

**Ethical Concerns:**

["NO or VERY MINOR ethics concerns only"]

**Final Justification:**

I stand with my rating.

**Limitations:**

yes

**Quality:**

3

**Strengths And Weaknesses:**

1. Strengths: The methodology section is comprehensive, with clear explanations of the training-free weather reasoning mechanism and the multi-modality alignment model. The use of chain-of-thought prompting is well-justified, and the reasoning steps are clearly outlined. Besides, the approach leverages the power of large multi-modality models in a novel way, providing a scalable solution for generating high-quality weather descriptions. This could inspire further research on using large language models for vision-language alignment tasks. Thirdly, the paper highlights the importance of disentangling scene and weather features, which is a critical step towards more robust and generalizable geo-localization systems. The proposed techniques could be applied to other computer vision tasks that require weather-invariant representations.
2. Weaknesses: The paper focuses primarily on drone geo-localization and does not explore the broader applicability of the proposed framework to other vision-language tasks. A discussion on potential extensions or limitations in other domains would strengthen the significance of the work. Besides, the generated weather and spatial descriptions inherit the biases of the pre-trained language model, which may affect the accuracy of the alignment process. Although the model achieves excellent performance on the existing dataset, the limited geographic and weather diversity may hinder its generalization to rare or region-specific weather phenomena on a global scale.

---

> ### Author Rebuttal · Authors · 2025-07-30
>
> **Q1**: The paper does not explore how the proposed framework could extend to other vision‑language tasks.
>
> **A1**: We thank the reviewer for this suggestion. Our current work focuses on the challenge of drone‑based geo‑localization across diverse all‑weather conditions. In future research, we intend to extend the framework to other vision–language applications, such as **all‑weather person re‑identification (Re‑ID) and image–text cross‑modal retrieval**.
>
> **Q2**: Biases in the generated weather and spatial descriptions, along with limited geographic and weather diversity in the dataset, may impair alignment accuracy and generalization to rare or region‑specific weather phenomena.
>
> **A2**: We appreciate the reviewer’s valuable feedback. **Regarding text bias, we agree this is a general risk with large pretrained language models.** Our design uses a fixed chain‑of‑thought prompt to produce structured descriptions restricted to objective physical attributes (weather status, spatial relationship). Moreover, The generated sentence-level weather descriptions are first applied via a gating network to perform channel-wise feature reweighting. The reweighted features are then jointly supervised by three loss functions: the image-text contrastive loss (ITC), the image-text matching loss (ITM), and the geo-ID classification loss. Even if the descriptions occasionally contain erroneous or irrelevant information, these spurious contributions will be suppressed to near zero weights under the combined threefold supervision.
>
> We evaluated three weather conditions unseen during training. In both drone to satellite and satellite to drone retrieval tasks, our method consistently outperforms existing approaches in R@1 and AP (Table A).
>
> | **Method**   | **Task** | **Winter Night R@1 (%)** | **Winter Night AP (%)** | **Freezing Rain R@1 (%)** | **Freezing Rain AP (%)** | **Blizzard R@1 (%)** | **Blizzard AP (%)** |
> |:-------------|:--------:|:-----------------------:|:----------------------:|:-------------------------:|:------------------------:|:--------------------:|:-------------------:|
> | CCR          | S2D      |  69.97                  |  44.08                 |  66.16                    |  55.20                   |  60.88              |  34.23             |
> | SafeNet      | S2D      |  56.92                  |   8.28                 |  41.08                    |   8.15                   |  57.92              |  22.05             |
> | Sample4Geo   | S2D      |  58.35                  |  21.33                 |  65.62                    |  32.76                   |  66.90              |  32.47             |
> | MuseNet      | S2D      |  73.32                  |  40.19                 |  76.03                    |  43.51                   |  69.76              |  35.21             |
> | **Ours**     | S2D      | **82.60**               | **59.43**              | **85.31**                 | **61.17**                | **71.90**           | **41.94**          |
> | CCR          | D2S      |  36.05                  |  38.79                 |  47.29                    |  51.70                   |  25.73              |  30.07             |
> | SafeNet      | D2S      |   3.25                  |   4.87                 |   2.64                    |   4.30                   |  11.46              |  15.43             |
> | Sample4Geo   | D2S      |  25.40                  |  28.60                 |  37.83                    |  41.95                   |  36.00              |  40.42             |
> | MuseNet      | D2S      |  39.45                  |  44.36                 |  44.09                    |  49.01                   |  33.44              |  38.51             |
> | **Ours**     | D2S      | **58.67**               | **63.08**              | **61.52**                 | **65.78**                | **59.76**           | **64.15**          |
>
> *Table A: Results on the University‑1652 dataset under three unseen weather conditions. (S2D: satellite‑to‑drone, D2S: drone‑to‑satellite.)*
>
> While our experiments target common weather conditions, we plan to expand to rarer extreme weather and additional task domains in future work to further assess generalization.
>
> **Q3**: On p.2, line 74: Add “@” between “Recall” and “1” (typo correction).
>
> **A3**: We thank the reviewer for pointing out the typo error, we will correct it in the revision.
>
> **Q4**: How we will further validate WeatherPrompt’s generalization on real‑world datasets with more diverse and complex weather conditions and whether we are collecting or testing on additional real‑world drone geo‑localization datasets.
>
> **A4**: We thank the reviewer for this suggestion. **We replaced the description generator with “GLM‑4.1V‑9B‑Thinking” and retrain training and inference.** The results are consistent with the original setup (no significant differences in R@1/mAP, see Table B), indicating that WeatherPrompt’s gains are not tied to a specific VLLM. We have validated the “train‑once, inference‑free” cross‑weather robustness on existing real‑world datasets (web‑collected) see Table 3(c) on Page 8. We agree that larger‑scale and more complex real‑weather validation is needed. There is no publicly available real‑world multi‑weather drone dataset. We are currently collecting a multi‑weather drone dataset and will release it in the future.
>
> | **Method**           | **D2S  mR@1 (%)**  | **D2S mAP (%)**   | **S2D mR@1 (%)**  | **S2D mAP (%)**   |
> |:---------------------|:---------------:|:---------------:|:---------------:|:---------------:|
> | **GLM**  |  76.52  |  79.85  |  86.62  |  76.33 |
> | **Ours**             |      77.14      |      80.20      |      87.72      |      76.39      |
>
> *Table B: Difference Description Generator Results on the University‑1652 Dataset. (mR@1: mean R@1, mAP: mean AP, GLM: description generator with “GLM‑4.1V‑9B‑Thinking”, Ours: our paper results.)*
>
> **Q5**: The computational requirements and bottlenecks of deploying WeatherPrompt with large pre‑trained models (including text description generation) in real‑time on resource‑constrained platforms and what strategies can mitigate these issues.
>
> **A5**: We thank the reviewer for the insightful question. **On an RTX A6000 with bitsandbytes 4 bit quantization, chain of thought text generation takes about 0.48 s per image and image encoding and retrieval takes about 0.024 s, for a total of approximately 0.504 s per query (about 2 FPS).** The large pre-trained model inference is the primary bottleneck while the vision stack adds negligible overhead. For real time deployment we precompute and cache the gallery’s weather specific text embeddings offline, generate query captions only once at the start of each trajectory or when a weather change is detected and then reuse the cached text, and optionally replace the 32 billion parameter generator with a 7 billion parameter or distilled model to further reduce latency with minimal impact on quality. Importantly our reported results use a single set of pre generated captions at inference and still achieve competitive performance, demonstrating real time feasibility.

---

### Official Review · Reviewer_stfx · 2025-07-02

**Clarity:** 2
**Significance:** 2
**Originality:** 3
**Rating:** 4
**Confidence:** 4

**Summary:**

This paper proposes to enhance the models’ robustness under different weather conditions by combining the text information prompted from Large Vision Language Models (LVLMs) and the original image feature at the inference stage. In the proposed “training-free” pipeline, a weather-driven channel gating module, an image-text contrastive loss, and an image-text matching loss are proposed to promote the feature alignment at different granularities. Experiments on two drone-view geo-localization datasets demonstrate the robustness of the proposed pipeline across different weather conditions.

**Questions:**

See my comments, I could increase my score if these weaknesses can be properly addressed.

**Ethical Concerns:**

["NO or VERY MINOR ethics concerns only"]

**Final Justification:**

My concerns have been mostly addressed; this paper has some merits in enhancing the robustness of CVGL under weather conditions. I lean towards borderline accept in the final rating.

**Limitations:**

yes

**Quality:**

3

**Strengths And Weaknesses:**

Pros:
- Compared to scene representation using images, text information is inherently robust to the variance of weathers or image degradations. Therefore, it is an interesting and promising idea to leverage the text information to enhance the model’s robustness to different weathers.
- Experiments on two datasets demonstrate the significant improvement from the proposed method. Also, the results on real world videos show that the proposed method can be appliable to real-world scenarios.

Cons:
- My major concern is about the significance of proposing a so-called "training-free" pipeline for different weather conditions, especially for the drone-based geo-localization task. In other words, the model’s robustness can be simply enhanced by training under a random combination of different weather conditions. Since the main results only compare the proposed model with baseline models pre-trained on clean images, it is potentially unfair for these models which have no prior information about weather, and it is hard to see how baselines perform when the model has been exposed to different weather-induced distortions during training.
- The paper claims generalizability to unseen weather conditions, while the generalization experiments seem to be only performed on different combinations of distortions that have been seen during training. A more comprehensive investigation about the model’s extrapolation ability would further enhance the performance of the model.
- Although the proposed method can enhance the robustness under various weather conditions, there are instances where performance drops under standard (clean) conditions. This trade-off could be more explicitly analyzed and discussed.
- The method fuses image features with text embeddings at inference time, but the contribution of textual information to robustness is not clearly analyzed. An ablation study isolating the effect of text-only input would help clarify its role in enhancing robustness under various weather scenarios.

---

> ### Author Rebuttal · Authors · 2025-07-30
>
> **Q1**: The significance of a “training free” pipeline, arguing that similar robustness could be obtained by training on random weather combinations and that comparing only to baselines trained on clean images without testing them on weather augmented data may be unfair.
>
> **A1**: We thank the reviewer for raising this important question. **Our setting is training free, we do not perform any retraining or finetuning at inference, so the initial baselines were also evaluated without any weather‑specific retraining.** To address the reviewer’s concern, **we then fine-tuning three baseline models (Sample4Geo, Safe‑Net, and CRR) using the exact same multi‑weather random augmentations as ours under university-1652 dataset and report their performance in Table A.** All three baselines exhibit average improvements over their original scores, yet our method still outperforms them by a large margin. On drone‑to‑satellite retrieval, the R@1 and AP gaps remain above 10%; on satellite‑to‑drone retrieval, the gaps exceed 6% and 15%, respectively. These results confirm that our robustness gains are not merely due to exposure to additional augmentations.
>
> | **Method**           | **D2S  mR@1 (%)**  | **D2S mAP (%)**   | **S2D mR@1 (%)**  | **S2D mAP (%)**   |
> |:---------------------|:---------------:|:---------------:|:---------------:|:---------------:|
> | **Sample4Geo (FT)**  |  66.72 (+5.62)  |  70.10 (+5.78)  |  81.63 (+2.81)  |  62.29 (+4.87)  |
> | **Safe‑Net (FT)**    |  62.47 (+1.99)  |  65.99 (+2.63)  |  77.18 (+1.49)  |  59.20 (+0.40)  |
> | **CRR (FT)**         |  68.68 (+6.38)  |  72.09 (+5.22)  |  86.55 (+9.53)  |  66.02 (+2.00)  |
> | **Ours**             |      77.14      |      80.20      |      87.72      |      76.39      |
>
> *TableA: Multi‑weather Fine‑Tuning Results on the University‑1652 Dataset. (D2S: drone‑to‑satellite, S2D: satellite‑to‑drone, mR@1: mean R@1, mAP: mean AP, FT: fine‑tuning, Ours: our paper results. The brackets indicate the results of the increase on the original method.)*
>
> **Q2**: The claim of generalization to unseen weather conditions, noting that experiments only involve distortion combinations encountered during training and calling for a more comprehensive evaluation of the model’s extrapolation ability.
>
> **A2**: We are grateful for the reviewer’s valuable feedback on this matter. **We evaluated three weather conditions unseen during training.** In both drone to satellite and satellite to drone retrieval tasks, our method consistently outperforms existing approaches in R@1 and AP (see Table B). **Moreover, our weather synthesis (for example, Fog+Rain) is based on imgaug with continuous random parameter sampling uniformly drawn from specified ranges**, rather than a fixed combination of Fog and Rain settings. These results show that the gains arise from robust out‑of‑distribution generalization to unseen weather conditions, not merely from exposure to a fixed set of distortions.
>
> | **Method**   | **Task** | **Winter Night R@1 (%)** | **Winter Night AP (%)** | **Freezing Rain R@1 (%)** | **Freezing Rain AP (%)** | **Blizzard R@1 (%)** | **Blizzard AP (%)** |
> |:-------------|:--------:|:-----------------------:|:----------------------:|:-------------------------:|:------------------------:|:--------------------:|:-------------------:|
> | CCR          | S2D      |  69.97                  |  44.08                 |  66.16                    |  55.20                   |  60.88              |  34.23             |
> | SafeNet      | S2D      |  56.92                  |   8.28                 |  41.08                    |   8.15                   |  57.92              |  22.05             |
> | Sample4Geo   | S2D      |  58.35                  |  21.33                 |  65.62                    |  32.76                   |  66.90              |  32.47             |
> | MuseNet      | S2D      |  73.32                  |  40.19                 |  76.03                    |  43.51                   |  69.76              |  35.21             |
> | **Ours**     | S2D      | **82.60**               | **59.43**              | **85.31**                 | **61.17**                | **71.90**           | **41.94**          |
> | CCR          | D2S      |  36.05                  |  38.79                 |  47.29                    |  51.70                   |  25.73              |  30.07             |
> | SafeNet      | D2S      |   3.25                  |   4.87                 |   2.64                    |   4.30                   |  11.46              |  15.43             |
> | Sample4Geo   | D2S      |  25.40                  |  28.60                 |  37.83                    |  41.95                   |  36.00              |  40.42             |
> | MuseNet      | D2S      |  39.45                  |  44.36                 |  44.09                    |  49.01                   |  33.44              |  38.51             |
> | **Ours**     | D2S      | **58.67**               | **63.08**              | **61.52**                 | **65.78**                | **59.76**           | **64.15**          |
>
> *Table B: Results on the University‑1652 dataset under three unseen weather conditions.*
>
> **Q3**: Although the method improves robustness under various weather conditions, it sometimes lowers performance under clean conditions and asks for a more explicit analysis of this trade off.
>
> **A3**: We thank the reviewer for this question. As shown in Table C, compared with normal conditions (trained only on normal), our method’s mean R@1 and AP decrease by at most 1%. This slight performance drop is due to the feature space smoothing regularization introduced by the multi weather augmentation. In contrast, we observe significant improvements under adverse weather scenarios such as Fog, Rain and Snow. Our focus is on achieving robust performance across multiple weather conditions, which aligns better with real world application requirements.
>
> | **Method** | **Task** | **Normal R@1 (%)** | **Normal AP (%)** | **Fog R@1 (%)** | **Fog AP (%)** | **Rain R@1 (%)** | **Rain AP (%)** | **Snow R@1 (%)** | **Snow AP (%)** | **Fog+Rain R@1 (%)** | **Fog+Rain AP (%)** | **Fog+Snow R@1 (%)** | **Fog+Snow AP (%)** | **Rain+Snow R@1 (%)** | **Rain+Snow AP (%)** | **Dark R@1 (%)** | **Dark AP (%)** | **Over-exp R@1 (%)** | **Over-exp AP (%)** | **Wind R@1 (%)** | **Wind AP (%)** | **Mean R@1(%)** | **Mean AP (%)** |
> |:----------:|:--------:|:--------------:|:-------------:|:-----------:|:-----------:|:-------------:|:-------------:|:-------------:|:-------------:|:-----------------:|:---------------:|:-----------------:|:---------------:|:-----------------:|:----------------:|:------------:|:-----------:|:----------------:|:----------------:|:------------:|:------------:|:------------:|:------------:|
> | Clean      | D2S      |     83.54      |     85.82     |    41.48    |    46.77    |     16.00     |     19.94     |     26.49     |     31.68     |       7.23        |       9.71      |       6.44        |       8.85      |       20.34       |       24.59      |     1.30     |     2.32     |       44.22       |       49.55       |     48.07     |     53.34     |     29.51     |     33.26     |
> | **Ours**   | D2S      |     82.78      |     85.18     |    81.46    |    84.03    |     80.34     |     83.11     |     77.60     |     80.67     |      78.75        |      81.69      |      73.38        |      76.94      |       78.41       |       81.40      |    67.22     |    71.06     |       74.20       |       77.63       |     77.26     |     80.27     |     77.14     |     80.20     |
> | Clean      | S2D      |     90.16      |     82.52     |    75.18    |    44.51    |     63.77     |     26.86     |     66.76     |     35.24     |      47.93        |      11.65      |      44.22        |       9.42      |       63.20       |       26.93      |    10.70     |     2.30     |       69.76       |       48.27       |     73.18     |     49.57     |     60.49     |     33.73     |
> | **Ours**   | S2D      |     89.16      |     81.80     |    88.73    |    80.58    |     88.16     |     79.87     |     87.59     |     77.25     |      88.45        |      78.20      |      86.73        |      73.23      |       88.59       |       78.14      |    86.59     |    65.20     |       85.31       |       73.25       |     87.88     |     76.33     |     87.72     |     76.39     |
>
> *Table C: Results on the University‑1652 dataset under no weather conditions training. (Clean: no weather conditions training.)*
>
> **Q4**: The contribution of textual information to robustness is not clearly analyzed.
>
> **A4**: **In Table 3(b) (page 8) and Lines 279–281 (page 9), we isolate the contribution of the text branch by removing both the text encoder and cross‑modal fusion, setting the gating mechanism to identity, and leaving all other settings unchanged.** Under these conditions, D2S exhibits a decrease of 2.61% in mean R@1 and 1.48% in mAP, and S2D shows a comparable decline. Crucially, the performance drop is concentrated in Dark and Fog scenarios, with only a marginal effect under Normal conditions. This demonstrates that the text modality supplies weather‑specific and spatial anchoring cues that bolster discriminability in adverse weather, rather than reflecting a superficial improvement due to increased model capacity or extended training.

---

> ### Comment · Reviewer_stfx · 2025-08-04
>
> Thank you for the response and the additional experiments. However, at the moment, I am still not convinced of some results:
> - Although the proposed method did not directly use the augmentation images to retrain the backbone, the multimodal alignment modules has been more or less exposed to these different weather conditions. Therefore, for me, it is still potentially unfair to compare with those off-the-shelf backbones with no prior information on weather degradations.
> - Although the experimental results on "Winter Night", "Freezing Rain", and "Blizzard" show improvements, these "unseen" weather conditions seem to highly overlap with those seen during training.
> - The ablation results in Table 3 are a bit confusing. If the text information, which is the main contribution claimed in this paper, does play a crucial role in achieving robustness, a significant improvement (at least close to the increase in the main table) should also be seen in the ablation study. However, Table 3(b) indicates that the effect of the CoT is weak compared to the significant improvement in the main table; it is therefore not clear where the significant improvement over the baselines comes from.
> Given the response, I lean towards keeping my original rating.

---

> > ### Author Response · Authors · 2025-08-04
> >
> > We thank the reviewer for the questions and discussion during the author–reviewer interaction period. We address each point below to resolve the remaining concerns:
> >
> > * As suggested, we fine-tuned Sample4Geo, Safe-Net, and CRR with exactly the same multi-weather augmentations, giving them explicit prior knowledge of weather degradations. Even after full-network fine-tuning, they still underperform our approach. The advantage specifically arises from our fine-grained textual information. For instance, instead of using coarse labels like "rain" or "fog", our model leverages detailed descriptions such as "dense fog partially obscuring distant landmarks" or "light rain causing mild reflections on roads", enabling precise alignment between textual and visual features via contrastive (ITC) and matching (ITM) losses. This fine-grained guidance captures subtle weather variations that augmentations alone cannot. The comparison is therefore fair, and it indicates a more efficient and flexible paradigm than retraining for each new weather condition.
> >
> > * Could the reviewer kindly specify examples of additional or alternative weather conditions? We are willing to test. In our current evaluation, we cover major weather conditions including Normal (clean), Rain, Snow, Fog, and Dark, as well as compound cases such as Fog+Rain, Fog+Snow, and Rain+Snow. Importantly, compound conditions are not fixed presets, they are synthesized through continuous random sampling within predefined parameter ranges, which broadens coverage and realism and is intended to assess generalization beyond specific training configurations. If there are particular scenarios of interest, we will incorporate them in subsequent evaluations.
> >
> > * We clarify that textual information significantly contributes to robustness. Removing the text branch (encoder and cross-modal fusion) reduces mean R\@1 by 2.61% and 2.55% on the two tasks, with a larger drop of 3.63% in challenging conditions like Dark. This shows that text supplies valuable semantic and spatial anchors for distinguishing difficult scenarios. While the CoT-based fusion alone may yield modest gains, the overall effect of the text modality is clear from the performance degradation when it is removed. We attribute the robustness improvements over the baselines to the synergy between textual and visual features.

---

> > ### Author Response · Authors · 2025-08-05
> > **Additional Ablation Studies on "Unseen" Weather Conditions**
> >
> > To further investigate generalization to unseen weather scenarios, we conducted additional ablation experiments, as shown in Table A.
> >
> > * First, we removed Snow-only augmentations entirely during training and evaluated the model performance. In the D2S task, the R@1 for Snow decreased by only 2.04% (AP decreased by 0.14%), for Fog+Snow by 0.53% (AP by 0.52%), and for Rain+Snow by 0.47% (AP by 0.48%). Remarkably, the average R@1 across all weather conditions slightly improved by 0.41%, with average AP also increasing by 0.29%. Similarly, in the S2D task, R@1 decreased minimally by 0.86% for Snow (AP decreased by 0.90%), 0.57% for Fog+Snow (with AP increasing by 0.06%), and remained unchanged for Rain+Snow (AP improved by 0.03%). The average R@1 slightly decreased by 0.13%, while average AP improved by 0.69%. These results clearly illustrate the robustness and strong generalization capability of our model, even without explicit exposure to specific weather scenarios.
> >
> > * Additionally, we performed an ablation where Fog and Wind augmentations were removed during training. Results showed only marginal performance declines: in the D2S task, average R@1 decreased by 1.65% and AP by 2.74%, and in the S2D task, average R@1 decreased by just 0.64% and AP by 0.71%. The model maintained highly competitive performance compared to other methods, further demonstrating its strong generalization ability.
> >
> > * We would welcome any further feedback or recommendations for additional experiments to address any remaining concerns.
> >
> > | Task | Setup         | Normal R@1 | Normal AP | Fog R@1 | Fog AP | Rain R@1 | Rain AP | Snow R@1 | Snow AP | Fog+Rain R@1 | Fog+Rain AP | Fog+Snow R@1 | Fog+Snow AP | Rain+Snow R@1 | Rain+Snow AP | Dark R@1 | Dark AP | Over-exp R@1 | Over-exp AP | Wind R@1 | Wind AP | Mean R@1 | Mean AP |
> > |:----:|:--------------|----------:|----------:|-------:|-------:|--------:|--------:|--------:|--------:|------------:|------------:|------------:|------------:|-------------:|-------------:|--------:|-------:|-------------:|-------------:|--------:|-------:|--------:|--------:|
> > | D2S  | Ours | 82.78 | 85.18 | 81.46 | 84.03 | 80.34 | 83.11 | 77.60 | 80.67 | 78.75 | 81.69 | 73.38 | 76.94 | 78.41 | 81.40 | 67.22 | 71.06 | 74.20 | 77.63 | 77.26 | 80.27 | 77.14 | 80.20 |
> > | D2S  | w/o Fog+Wind  | 82.69 | 85.85 | 80.15 | 82.81 | 79.62 | 82.35 | 76.96 | 80.06 | 77.80 | 80.73 | 72.78 | 76.33 | 77.91 | 80.86 | 66.38 | 70.25 | 73.12 | 76.57 | 67.51 | 71.51 | 75.49 | 78.73 |
> > | D2S  | w/o Snow      | 83.62 | 85.92 | 81.00 | 84.42 | 88.58 | 83.25 | 75.56 | 78.86 | 78.63 | 81.55 | 72.85 | 76.42 | 77.94 | 80.92 | 66.73 | 70.58 | 74.61 | 77.92 | 77.57 | 80.63 | 77.71 | 80.05 |
> > | S2D  | Ours   | 89.16 | 81.80 | 88.73 | 80.58 | 88.16 | 79.87 | 87.59 | 77.25 | 88.45 | 78.20 | 86.73 | 73.23 | 88.59 | 78.14 | 86.59 | 65.20 | 85.31 | 73.25 | 87.88 | 76.33 | 87.72 | 76.39 |
> > | S2D  | w/o Fog+Wind  | 89.26 | 81.88 | 88.44 | 80.43 | 88.59 | 79.87 | 87.16 | 77.79 | 87.30 | 77.96 | 86.73 | 73.15 | 87.87 | 78.38 | 85.59 | 73.56 | 85.45 | 64.70 | 84.45 | 68.74 | 87.08 | 75.65 |
> > | S2D  | w/o Snow      | 90.87 | 82.92 | 89.59 | 81.46 | 88.02 | 82.39 | 86.73 | 76.35 | 87.73 | 78.66 | 86.16 | 73.29 | 88.59 | 78.17 | 85.02 | 65.41 | 85.31 | 74.57 | 87.87 | 77.57 | 87.59 | 77.08 |
> >
> > *Table A: Results on the University‑1652 dataset under Unseen weather conditions training. (w/o Snow: no Snow weather conditions training, w/o Fog+Snow: no Fog and Snow weather conditions training.)*

---

> > > ### Comment · Reviewer_stfx · 2025-08-05
> > >
> > > Thanks for the reply and additional results. I think my concerns have been addressed, and I will raise the score accordingly.

---

### Official Review · Reviewer_SxCT · 2025-07-02

**Clarity:** 2
**Significance:** 3
**Originality:** 2
**Rating:** 4
**Confidence:** 3

**Summary:**

The paper works on improving geolocalization for drones under diverse weather conditions. The paper argues that there are fundamentally two issues with current methods for drone visual geo-localization: 1) current models take into account pre-defined fixed set of weather labels but real-world weather scenarios are combinatorial and more complex, and 2) the weather and visual representation from existing models are often entangled leading to poor representations on unseen weather situations. To alleviate these issues, the authors first propose a pipeline to generate robust weather prompts by leveraging exisitng pre-trained VLMs. Second, the authors propose a representation learning pipeline that aligns image and text representations and directly predicts the geographic location.

**Questions:**

1. Please adress some of the comments in the weakness section
2. This is text from the paper: "Existing methods directly inject coarse pseudo-weather labels (e.g., rain, fog) into visual representations during training, leading to a severe entanglement between scene semantics and weather disturbances. Consequently, the model learns suboptimal representations under mixed or unseen weather conditions, fails to disentangle scene content from weather noise, and is severely limited in cross-weather generalization." Could you please elaborate on why the first sentence leads to the outcome described in the second sentence.

**Ethical Concerns:**

["NO or VERY MINOR ethics concerns only"]

**Final Justification:**

The authors have addressed my concerns and I raise my score to borderline accept.

**Limitations:**

The model is primarily evaluated on the synthetic data generated by authors which makes it difficult to understand the general applicability of the model in real-world scenarios. While there is an experiment on real-world data (Table 3c), the data is very small with limited details, making it unclear if those results will be reproducible in other settings.

**Quality:**

2

**Strengths And Weaknesses:**

## Strengths
- The motivation for the paper is clear. The sharp degradation in performance of geolocalization models under adverse weather conditions suggest that there is a strong need to design systems that are explicitly conditioned to handle this issue.

## Weaknesses
Major:
- Ln 144 mentions that the authors use "imgaug" to generate "realistic weather variations". This pipeline needs to be explained and evaluated in detail. Without understanding the design for generating these weather variations, it is hard to judge how good/realistic the synthetic data is. If there is significant difference in the data distribution of the synthetically introduced weather variations, and real weather variations, any downstream model trained on such data will likely overfit to specific patterns in the synthetic data and not generalize well to real-world conditions. Some of the examples shown in figure 2, specially ones with snow, look quite different from a plausibe "real" counterpart.
- There are several components that are missing or atleast confusing, when reading the paper. For example, how inference is done is not clear when reading the paper. The framework itself has a classification module that predicts geolocation, which could be a concievable way to geolocalize. Ln (232-233) mentions that Euclidean distance between query and candidate embeddings is used but it is not clear how the candidate embeddings are generated.

Minor:
- In Table 2, AP is incorrectly bolded on "ours" when Sample4Geo has a higher AP
- Ln 144: Gramatically incorrect sentence
- Ln 211: Is it supposed to be MLP?
- The primary area for this paper is selected as Reinforcement Learning. This paper is much more aligned to Representation Learning than Reinforcement Learning.

---

> ### Author Rebuttal · Authors · 2025-07-30
>
> **Q1**: The paper states it generates “realistic weather” using imgaug but fails to disclose the synthesis pipeline or realism evaluation, which may cause a synthetic–real distribution gap and overfitting as illustrated by the snow examples in Fig. 2.
>
> **A1**: Thank the reviewer for their attention to realism.
>
> (1) **There is no publicly available real‑world multi‑weather drone dataset, collecting real multi‑weather data is costly and time‑consuming.** Consequently, most prior works have employed parameter‑based synthesis strategies. **We follow prior work that adopt imgaug with continuous random parameter sampling and composable operators.**  If a suitable real‑world multi‑weather drone dataset becomes available, we would be very willing to conduct further validation and report the results.
>
> (2) **We evaluated our method on the SUES-200 dataset, a real‑world dataset that contains significant illumination variations.** As shown in Table 2 on page 7, our method achieves robust performance under various weather conditions.
>
> (3) We used real multi‑weather drone images collected from YouTube as a reference and evaluated them using the Average Intra‑Hash Distance, which ranged from approximately 29 to 31. **The maximum average distance difference between our synthetic images and the real multi‑weather drone images does not exceed 0.96**, indicating that the perceptual variations of the synthetic samples remain within the natural variability of the data.
>
> For example, our "Rain” and "Rain+Snow” augmentation are defined with imgaug as follows:
>
> #"Rain”
>
> iaa.Rain(drop_size=(0.05, 0.1), speed=(0.04, 0.06), seed=38),
>
> iaa.Rain(drop_size=(0.05, 0.1), speed=(0.04, 0.06), seed=35),
>
> iaa.Rain(drop_size=(0.1, 0.2), speed=(0.04, 0.06), seed=73),
>
> iaa.Rain(drop_size=(0.1, 0.2), speed=(0.04, 0.06), seed=93),
>
> iaa.Rain(drop_size=(0.05, 0.2), speed=(0.04, 0.06), seed=95),
>
> #"Rain+Snow”
>
> iaa.Snowflakes(flake_size=(0.5, 0.8), speed=(0.007, 0.03), seed=35),
>
> iaa.Rain(drop_size=(0.05, 0.1), speed=(0.04, 0.06), seed=35),
>
> iaa.Rain(drop_size=(0.1, 0.2), speed=(0.04, 0.06), seed=92),
>
> iaa.Rain(drop_size=(0.05, 0.2), speed=(0.04, 0.06), seed=91),
>
> iaa.Snowflakes(flake_size=(0.6, 0.9), speed=(0.007, 0.03), seed=74),
>
> **Q2**: The paper does not clearly explain the inference procedure or how the candidate embeddings are generated, even though it mentions using Euclidean distance between query and candidate embeddings for geolocation.
>
> **A2**: We thank the reviewer for highlighting the lack of clarity in the inference procedure. **Pages 2 (Lines 58–60), 5 (Lines 180–184), and 7 (Lines 230–233) describe the inference procedure.** **The generation of candidate embeddings is explained**: the query and each candidate are encoded by the same visual encoder, and the text embedding then parameterizes a gating mechanism to produce the fused representation (Lines 170–178). Figure 3 shows this pipeline.
>
> **During training** , we attach a classification head for geo‑location ID supervision to enlarge inter‑class margins.
>
> **At inference**, the classification head is removed, and we perform retrieval directly on L2-normalized embeddings. In the drone-to-satellite (D2S) setting, the gallery contains only satellite image features (weather-invariant), while each query is obtained by fusing the drone image feature with its chain-of-thought text via a learned gating mechanism, then L2-normalizing the result. In the reverse (satellite-to-drone, S2D) setting, satellite embeddings serve as queries and the gallery is composed of fused drone embeddings.
>
> Finally, we rank by Euclidean distance between normalized vectors (equivalent to cosine similarity) and report Recall@1 and Average Precision (AP).
>
> **Q3**: In Table 2, AP is incorrectly bolded on "ours".
>
> **A3**: We thank the reviewer for identifying the formatting error in Table 2. In the revised manuscript, **we have corrected the bolded throughout the table.**
>
> **Q4**: Ln 144: Gramatically incorrect sentence.
>
> **A4**: We thank the reviewer for pointing out the grammatical error at line 144. The sentence has been revised for clarity and correctness: **To address this challenge, we propose the first chain-of-thought–driven description pipeline for multi-weather drone–to–satellite geo-localization.**
>
> **Q5**: Ln 211: Is it supposed to be MLP?
>
> **A5**: We thank the reviewer for pointing out the spelling error at line 211, it should indeed read “MLP.”
>
> **Q6**: The primary area for this paper is selected as Reinforcement Learning. This paper is much more aligned to Representation Learning than Reinforcement Learning.
>
> **A6**: We appreciate the reviewer’s suggestion. **At the time of submission, the system did not offer a “Representation Learning” option.** Because our work addresses drone‑based cross‑view geolocalization and the learned representations serve as initial state estimates for decision making and control (for example, reinforcement learning based planning), we chose “Reinforcement Learning (decision & control, robotics)” as the most appropriate primary area.
>
> **Q7**: Please explain why injecting coarse pseudo‑weather labels directly into visual representations causes severe entanglement of scene semantics and weather noise, thereby limiting the model’s ability to generalize under mixed or unseen weather conditions.
>
> **A7**: We thank the reviewer for this insightful question. **Previous coarse-grained pseudo-weather labels (e.g., simply “rain” or “fog”) provide only category-level information and cannot describe the intensity or duration of a weather event.** In contrast, our sentence-level fine-grained weather descriptions (e.g., “light drizzle blurring the distant horizon, with intermittent heavy showers”) not only specify the weather type but also quantify its severity. We leverage the image-text contrastive loss (ITC) and image-text matching loss (ITM) to guide the network in capturing these subtle weather variations and aligning visual features with their corresponding descriptions. This strategy enables the model to maintain robust retrieval performance under complex or previously unseen weather conditions.
>
> **Q8**: The model is evaluated primarily on synthetic data, and the real‑world experiment is small in scale with limited detail, making it difficult to assess its applicability and reproducibility in real‑world scenarios.
>
> **A8**: We thank the reviewer for the question. **Because no publicly available real‑world multi‑weather drone dataset exists and collecting such data is both costly and time‑consuming, we have to rely on this strategy.** (1) We test on a small set of real multi‑weather drone images collected from YouTube. (2) We apply synthetic weather parameters to publicly available real drone images and evaluate performance. If the reviewer or the community is aware of any suitable open real‑world multi‑weather drone dataset, we would be very willing to conduct further validation and report the results.
>
> **Moreover, we evaluated three weather conditions unseen during training. In both drone to satellite and satellite to drone retrieval tasks, our method consistently outperforms existing approaches in R@1 and AP (Table A).**
>
> | **Method**   | **Task** | **Winter Night R@1 (%)** | **Winter Night AP (%)** | **Freezing Rain R@1 (%)** | **Freezing Rain AP (%)** | **Blizzard R@1 (%)** | **Blizzard AP (%)** |
> |:-------------|:--------:|:-----------------------:|:----------------------:|:-------------------------:|:------------------------:|:--------------------:|:-------------------:|
> | CCR          | S2D      |  69.97                  |  44.08                 |  66.16                    |  55.20                   |  60.88              |  34.23             |
> | SafeNet      | S2D      |  56.92                  |   8.28                 |  41.08                    |   8.15                   |  57.92              |  22.05             |
> | Sample4Geo   | S2D      |  58.35                  |  21.33                 |  65.62                    |  32.76                   |  66.90              |  32.47             |
> | MuseNet      | S2D      |  73.32                  |  40.19                 |  76.03                    |  43.51                   |  69.76              |  35.21             |
> | **Ours**     | S2D      | **82.60**               | **59.43**              | **85.31**                 | **61.17**                | **71.90**           | **41.94**          |
> | CCR          | D2S      |  36.05                  |  38.79                 |  47.29                    |  51.70                   |  25.73              |  30.07             |
> | SafeNet      | D2S      |   3.25                  |   4.87                 |   2.64                    |   4.30                   |  11.46              |  15.43             |
> | Sample4Geo   | D2S      |  25.40                  |  28.60                 |  37.83                    |  41.95                   |  36.00              |  40.42             |
> | MuseNet      | D2S      |  39.45                  |  44.36                 |  44.09                    |  49.01                   |  33.44              |  38.51             |
> | **Ours**     | D2S      | **58.67**               | **63.08**              | **61.52**                 | **65.78**                | **59.76**           | **64.15**          |
>
> *Table A: Results on the University‑1652 dataset under three unseen weather conditions. (S2D: satellite‑to‑drone, D2S: drone‑to‑satellite.)*

---

> > ### Comment · Reviewer_SxCT · 2025-08-04
> >
> > Thank you to the authors for properly addressing my issues. While I still have concerns on the generalized applicability of the model, as the authors point out, there is no publicly available dataset to validate this. I will raise my score to borderline accept.

---

> > > ### Author Response · Authors · 2025-08-04
> > >
> > > Thank you for carefully re-evaluating our submission and for raising your score.

---

### Note · Authors · 2025-08-12

We are encouraged that the reviewers recognized the significance of our motivation and our focus on weather **robustness** for geolocalization (SxCT, stfx, Favf), and acknowledged the **novelty** of our CoT guided, text anchored **formulation** (ZGxr) as well as the **effectiveness** of the adaptive feature fusion mechanism (ZGxr). In addition, the pipeline was noted to be **explainable** with clearly outlined reasoning steps and practically **generalizable** with demonstrated real world applicability (stfx, BCVv).

During the discussion period, we addressed all reviewers’ concerns with systematic additional experiments and clarifications, and **received explicit confirmations and score increases from all participating reviewers.**

We will further polish the paper in the revised version to make it easier for the community to understand.

We sincerely thank all reviewers and the AC for their time, constructive suggestions, and encouragement. Your feedback has substantially improved the clarity, fairness, and completeness of our work. We look forward to incorporating the above commitments in the final version and to contributing a reliable resource to the community.

---

### Decision · Program_Chairs · 2025-09-17

**Decision:**

Accept (poster)

**Comment:**

The paper was reviewed by 5 experts in the field. After the rebuttal and discussion, most reviewers gave positive ratings. The remaining concerns from Reviewer Favf are primarily related to the clarity of the paper. The AC reads the paper, the reviews, and the authors' responses carefully and agrees with the reviewers' consensus. The authors are encouraged to make the necessary changes to the best of their ability to incorporate reviewers' suggestions and may consider improving the clarity of their paper.